# Fragments for the History of an Architecture: A Villa between Humanism and the Renaissance

Camilla Mileto * and Fernando Vegas

Centro de Investigación en Arquitectura, Patrimonio y Gestión para el Desarrollo Sostenible (PEGASO),
Universitat Politècnica de València, 46022 Valencia, Spain; fvegas@cpa.upv.es
* Correspondence: cami2@cpa.upv.es

**Abstract:** This article presents a detailed study of the stately palace of the Villa Giusti-Puttini, a building that, over the centuries, has undergone repeated transformations since its construction in the first half of the 15th century. For the study of this palace, owned between the 15th and 17th centuries by one of the most important families in the city of Verona (Italy), the authors have followed a methodology covering indirect sources (documentary and bibliographical) as well as direct ones (the building and constructive techniques, architectural and decorative elements, murals, etc.). This study expands the information available on the building as well as expanding knowledge on the history of architecture of the Veneto villa as a defining architectural phenomenon in 16th- and 17th-century architecture whose extensive influence was still felt in the 19th century. The history compiled through this research also contributes to a renewed interpretation of the phenomenon, which is viewed as a process for the transformation and adaptation of a pre-existing building to fit the needs of any given period. This methodology, which could potentially facilitate the interpretation of similar buildings, and its combination of documentary, material, constructive, decorative, and cultural elements could constitute an example for the historical and architectural reading of buildings and are not merely limited to Renaissance buildings.

**Keywords:** history of architecture; Veneto villa; Verona; study methodology; chrono-typology; material history; Giusti family; frescoes; floors

## 1. Introduction

### 1.1. Villas in Veneto: A Cultural, Social, Economic, and Architectural Phenomenon

The Republic of Venice began its rise in the 9th century, reaching its apogee in the 14th century when its territory covered the east coast of the Adriatic Sea (Istria and Croatia) as far as Greece, incorporating most of the Adriatic islands (Figure 1a). It was at this point that the Republic began to focus on its mainland territories, which reached their maximum extent in successive campaigns in the space of a century (from the 1330s to the 1440s) [1]. The newly rediscovered peace in this land brought about a new phenomenon [2]: the wealthy families, both local and from Venice, began to invest in the countryside in the different provinces, and the Republic started to diversify its income thanks to the growing agricultural economy. In the late 16th century, as part of the process of conquering the mainland, the *villeggiatura,* the course of the summer period in the rural villa, offered the perfect opportunity for families, especially those from the Venetian nobility, to show off their economic power [3,4]. Villas became spaces for sumptuous parties and luxurious banquets, reaching unprecedented heights in the second half of the 17th century and throughout the 18th century, unrivalled in Europe and the rest of Italy. Furthermore, in the most remote territories of Venice, the development observed in the villas belonging to local families was different, as they continued to focus mostly on production while incorporating the literary humanistic ideal of the country house as "locus amoenus", that is, a refuge that was also offered as a meeting place for the literary and cultural circles of the time [5].

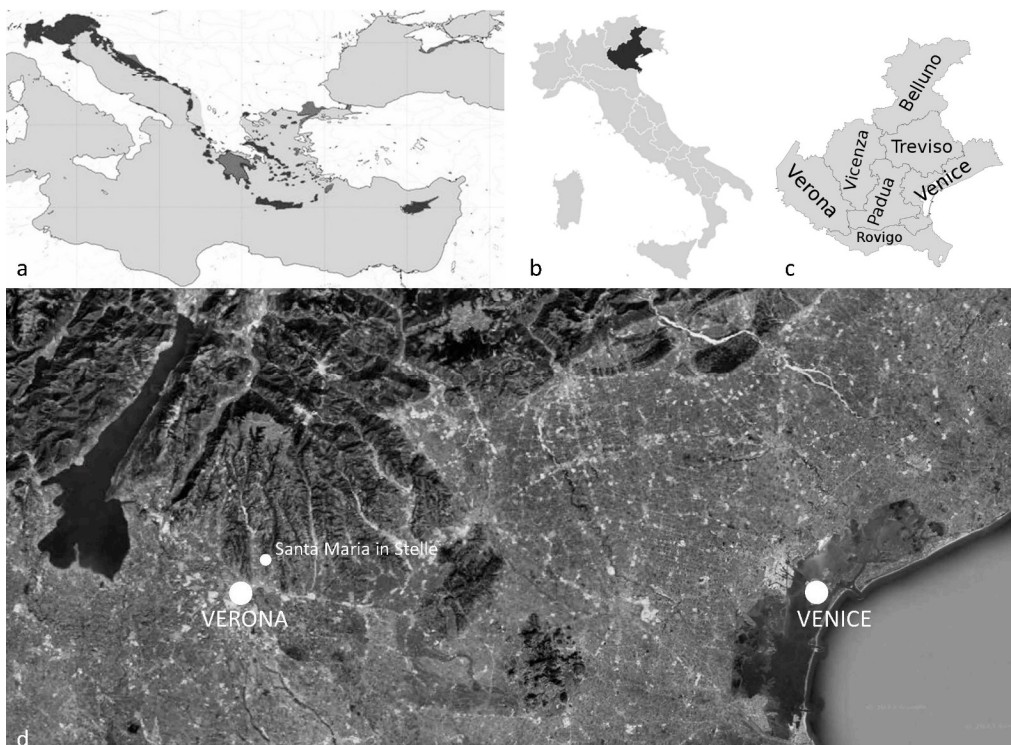

**Figure 1.** Location map: (**a**) The extension of the Republic of Venice throughout its history; (**b**) the location of the Veneto region today; (**c**) the situation of the provinces of the Veneto region at present; and (**d**) the location of the village of Santa Maria in Stelle in relation to the city of Verona.

In this regard, in the Veneto (Figure 1b), the villa was the complex made up of the main residence, where the owner lived, or palace, the adjoining land (fields, orchard, park, and garden), any buildings erected there (chapels, pergolas, mills, cisterns, aviaries, housing for sharecroppers and labourers, haystacks, haylofts, sheds, cool houses, and pools), and the men who worked and lived on the land (master, foreman, sharecroppers, labourers, domestic workers, and servants, with their respective families). The mostly productive nature of the Veneto villa took material form, from the earliest 15th century examples, as a stately palace strictly connected to its agricultural building annexes. The first complexes or villas in the 15th century in the Veneto region were not built by famous architects. These tended to be modest complexes whose raison d'être was to link agricultural functions and the master's country residence. The first representation of a villa complex was recorded in the treatise *De Agricoltura* by Pietro Crescenzi, written in the 14th century and published in the late 15th century [6–8]. Prior to this, the only examples of noble architecture in the countryside had been castles or castle-palaces. From the 15th century on, the initial process of adaptation of the existing buildings was followed by another of new construction, which led to the consolidation of the villas, reaching their apogee in the Palladian villas [9]. In this regard, pre-Palladian villas [10] took material form in a wide range of possibilities, which included tower-villas or castle-villas [11,12], rural houses adapted into stately palaces [13–16], and country palaces built to imitate city ones [17,18]. All of these tried to find solutions to guarantee the representativeness of these buildings within a productive rural setting. In fact, Palladian villas are clearly linked to this phenomenon [19], combining representative elements that are generally concentrated in the main body of the villa or palace with clearly functional elements derived from the productive needs of these complexes (dovecote tower, outbuildings for agricultural implements, grain stores, etc.) [19].

### 1.2. The Five Villas of the Giusti Family in Santa María in Stelle

A clear and relevant case for understanding the transformations that gradually took place in the architecture of villas on the mainland is that found in the village of Santa Maria

in Stelle and the neighbouring district of Vendri, in the province of Verona (Figure 1c,d). This group of five villas belonged to the same branch of the Giusti di San Quirico family, later Giusti delle Stelle. The Giusti family was one of the most important in the city of Verona in the 15th century. Family members are recorded in the city records as early as the 14th century, listed among wealthy wool merchants who had the right to take an active part in the city's political decisions. It was with Lelio Giusti that the history of this family was linked to a small village in the hills of Verona, Santa Maria in Stelle, following his marriage to Zilia Campagna, widow of Bartolomeo Montagna, in 1446. Through this marriage, Lelio acquired his first villa, consisting of the stately palace, the courtyard, the outbuildings, the dovecote, a garden, and agricultural land. This complex, which is now part of the villa Giusti-Puttini, analysed in depth in this text, marked the start of the Giusti family's association with the location, which would last for centuries and take material form in another four properties [19]: the palace of Zenovello Giusti, Corte ai Casai, villa Giusti-Bianchini, and villa Giusti-Melloni. Selected information on these four properties is included here to provide context to help understand the main subject of this text, Villa Giusti-Puttini, which will be described from the second section onwards.

The so-called Palace of Zenovello was built by Zenovello Giusti as part of the inheritance received following the death of his father, Lelio, in 1482. After years of conflict about the division of his father Lelio's inheritance, Giusto inherited the properties that had belonged to the Montagna family and that are currently known as Villa Giusti-Puttini. However, in the late 15th century (approx. 1490), his brother Zenovello used his share of the inheritance to build a sumptuous palace on the land immediately opposite Villa Giusti-Puttini. Zenovello died in 1503 and left his assets to his three children, stipulating that the palace of Santa Maria in Stelle should always be passed down to the eldest male child of the family. The floor plan of the palace and its gardens are recorded in detail in a plan dating from 1625, when it must still have been in use as a luxury dwelling for this branch of the Giusti family. In 1675, the Republic of Venice condemned Zenovello's heirs to exile for the kidnapping of Angela Lonardi [20,21] and ordered the expropriation of their assets, the loss of the aristocratic titles, and the destruction of the family palace in Santa Maria in Stelle [22], to the point that the Palace of Zenovello appears on a plan from 1683 as a complex in ruins described as a "Palazzo delle Stelle Distrutto". However, the 1625 floor plan suggests that it must have been a large residence with a tripartite floor plan, perhaps following the model of city palaces, with a large central chamber on the floor plan with representative functions [19]. On the plot that was attached to the palace, it is still possible to find some structures that incorporate incrusted remains both of the palace of Zenovello (fragments of cornices, windows, columns, etc.) and Roman remains that may possibly have been in the palace itself. In addition, it is highly probable that the walls of these buildings, some of which were built with pebbles, belonged to the former palace.

Linked to the history of the Palace of Zenovello, we find the Corte ai Casai, also located in the village of Santa Maria in Stelle, in the highest part, west of the current parish church. Over the centuries, this complex has undergone numerous transformations and adaptations, and, as stated in a note on the plan from 1735, it was known as "Casa domenicale detta delli Casali delli Sig.ri Co: Co: Gio: Battista Giusti e Cesare Nipote Goiusti habitata dalli Sud:ti Co:Co: dopo la distruzione del Pallazzo e migliorata". In this case, it is a credible hypothesis that, following the destruction of the Palace of Zenovello in 1675, the Corte ai Casai was renovated and adapted with elements and fragments from the palace for use by the family [23]. In fact, fragments from the palace can still be seen encrusted in the construction of modest service buildings located on the site of the ruined palace.

Opposite the village parish church, we find the Villa Giusti, now Bianchini. This longitudinal building is made up of two adjoining bodies: a quadrangular body built in the 18th century [24], and to the east, an elongated body with a portico built in the mid-16th century [25]. The latter, which incorporates a wide portico with masks on the ground floor and vaults with stucco in the interior, is attributed to Veronese artist Bartolomeo Ridolfi [25,26], cited by Palladio in his treatise as one of his repeated collaborators. The

longitudinal composition of this body with a portico is reminiscent of references from Renaissance cultured architecture (including Loggia della Regina Cornaro in Altivole, Loggia di Frà Giocondo in Verona, and Loggia di Alvise Cornaro in Padua, built by the Veronese Giovanni Maria Falconetto), as well as references to local constructive tradition in the buildings with porticos and loggias found in Verona and Vicenza [19].

The villa Giusti-Melloni, which long ago was the property of the same branch of the Giusti delle Stelle family, is located in the nearby village of Vendri. It probably dates from the second half of the 16th century [27] and first appeared on a plan from 1598, described as a "burned palace", due to a fire caused in 1585 by a member of the Giusti family seeking revenge. The villa features a complex made up of the palace with a square floor plan, the front courtyard, and the loggia, which is still used to access the complex, as well as a garden at the back. Above the three arches of the main façade, the crests of the Giusti family appear in the centre, with the crests of the women's families (Emilei and Pompei) on either side. These are all found within a monochrome fresco by Paolo Farinati (1524–1606) [27,28]. The composition of the floor plan and the height of the palace spaces are directly reminiscent of different Palladian villas, although the layout of the floor plans by Palladio is always much more articulated and varied than that found in this Villa Giusti.

## 2. Subject of Study, Objectives, and Methodology Used

### 2.1. The Subject of Study

As seen above, the Villa Giusti-Puttini (Figure 2) was the first property owned by the Giusti delle Stelle family, speaking in chronological terms. The fact that this complex has witnessed the passing of the centuries through different owners and situations has resulted in a case that provides the wealth of data to be studied in this article. The villa has been partially studied by several authors who have focused on its history [24], particularly on the decorative elements and frescoes covering the interior rooms [24,28,29]. Its architecture has also been studied by Camilla Mileto, co-author of this text, since the late 1990s [30] and successively expanded with new interpretations [19] up until this one, which aims to provide a detailed overview of the complexity of the villa and its evolution over time.

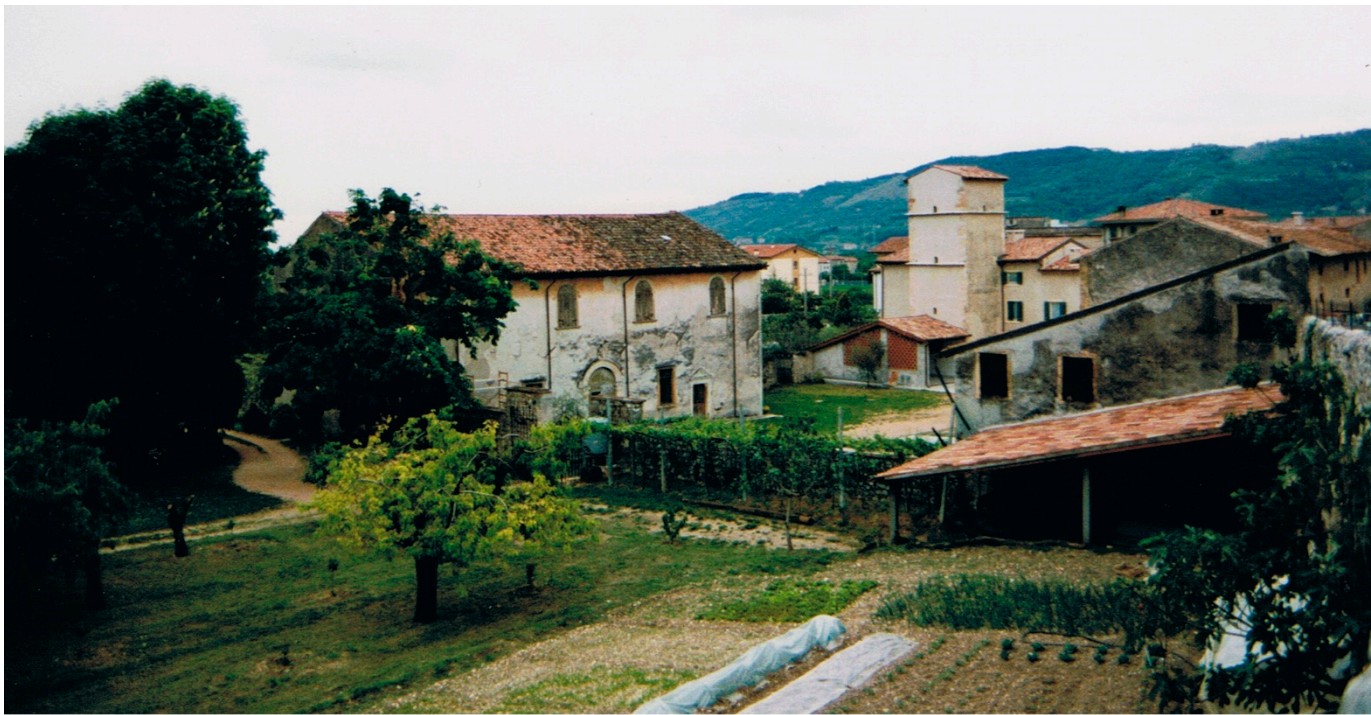

**Figure 2.** View of the complex of the Villa Giusti-Puttini in Santa María in Stelle (Verona).

*2.2. Study Methodology*

The study was carried out in different phases. The first [30] focused in depth on the building in historic, constructive, and structural terms; in the second [19], hypotheses were drawn up regarding the transformations of the building through stratigraphic interpretation and comparison with similar buildings; in the third phase, following an updated examination of the recent bibliography, new non-destructive tests (a thermographic study) and chrono-typological studies of architectural, constructive, and decorative elements were carried out, expanding knowledge of the building and helping develop more detailed hypotheses on its constructive phases. This unpublished research is recorded in this article.

The building was therefore studied using both indirect and direct sources and mostly non-destructive methods. The study of the indirect sources required extensive archival research both in the Archivio di Stato di Verona and in the Archivio di Stato di Venezia. This made it possible to locate the written documentation relating to wills, ownership transfers, inheritance documents, and valuations, together with plans and maps linked to the management of water and the properties, as well as 19th and 20th century cadastres. Thanks to these documents, it was possible to identify the owners of the villa over the centuries. At the same time, bibliographical research was carried out, both on the few sources devoted to the villa [23,24,28,29,31] and on those related to the Giusti family (including [32–38]), the villas of the Veneto (among the many possible references: [5–7,9,39,40] and on those of the province of Verona (including [17,27,41,42]) and on local artists and architects [40,43,44]. This study has made it possible to establish references and relationships between the villa and its context.

A detailed study was also carried out on different aspects of the architecture, construction, ornamentation, etc. An initial metric survey was completed to identify the measurements and the geometric deviations that the different building volumes have revealed. Secondly, the constructive materials and techniques of the constructions, renderings, floors, and ceilings were also studied. Particular attention has been paid to the study of the ceilings, as this complex of Renaissance structures was particularly well conserved. Thirdly, a stratigraphic study of the construction was completed to examine the stratigraphy of the architecture [45] and to help establish a relative chronology and order of phases for the construction. Fourthly, a study was carried out of the mechanisms of degradation of constructions, floors, and ceilings, as well as of the structural mechanisms and lesions of the building. Finally, in this first study phase, conservation and restoration actions were proposed for the conservation of the complex.

In 2008, the start of a second study phase was initiated, which aimed to complete the previous one by clarifying some unknowns with the help of thermography and increasing the accuracy of stratigraphic interpretations at some points. In the third study phase, in 2022 and 2023, a study and classification were carried out of all the decorative elements, including windows, doors, corners, mouldings, etc., for their comparative analysis and that of other similar elements found in the city of Verona. A more in-depth study was also carried out on the decorative motifs of the ceilings and the stylistic motifs of the mural paintings covering different rooms inside the palace. Along with the previous one, this more recent research phase has made it possible to expand knowledge of this unique complex. The results of these studies and the derived hypotheses are presented in the following section.

## 3. Results

*3.1. Fragments of Family Histories*

In broader terms, the history of the changes in ownership of the Villa Giusti-Puttini can be reconstructed thanks to historical documents and plans. The first document in which the villa is clearly mentioned is that relating to the contribution of the dowry of Zilia Campagna, widow of Bonsignorio Montagna, who went on to marry Lelio Giusti in 1446 (A.S.Vr, Archivietti Privati, Giusti, Arc. 34, vol. II; ff. 19r–22r). Zilia's inheritance was the result of the division of her first husband's estate between her and her husband's daughter,

carried out in 1445 after Bonsignorio's death (A.S.Vr, Archivietti Privati, Giusti, Arc. 34, vol. II; ff. 192v–205r). It is not known exactly how long the villa has been owned by the Montagna family, as can be seen from the crest inserted in the north façade of the palace. The Montagna were a family that had been established in Verona since the 13th century. In the 14th century, there were three family branches sharing the same crest (with a mountain with six peaks and some stars): the Montagna family in San Quirico, in San Martino Aquaro, and in San Paolo [41]. Zilia Campagna was married to Bonsignorio Montagna from San Quirico, while the branch of San Martino Aquaro owned property in San Zeno de Montagna. This branch of the family was particularly important, as it was there that the stately palace built in the first half of the 15th century was to be found [41]. Although these were two different branches of the family, they were probably connected [41]. In this respect, there was possibly some relation between this stately building, known as Casa Montagna, and the stately palace of the villa Giusti-Puttini. Casa Montagna, probably all that remains of what must have been a villa, has three levels and a ground floor portico leading onto two large rooms (possibly the stable and hayloft), an upper floor with a gallery with three windows, and a kitchen at the end, from which two large rooms and the grain store of the attic are accessed.

As mentioned above, with the marriage between Zilia Campagna and Lelio Giusti in 1446, the villa and the palace of San Quirico in Verona became the property of the Giusti family, wool merchants who had become increasingly powerful in city politics, with several members becoming important personalities. In fact, several authors [33,34] refer to Lelio as Podestá of Firenze. Following Lelio's death in 1482 and after a lengthy dispute, his sons Giusto and Zenovello divided up the inheritance between 1488 and 1490 (A.S.Vr, Archivietti Privati, Giusti, Arc. 34, vol. I; ff. 231r–213v) so that Giusto was assigned the "Casa nostra delle Stelle col Giardino", current Villa Giusti-Puttini, and Zenovello the "Casale del lago" perhaps the place where he started construction on what became known as the New Palace or Palace of Zenovello. Giusto, son of Lelio and married to the noblewoman Laura D'Arco, is remembered as a "dottore ed oratore eccellentissimo" and was a renowned humanist in Veronese cultural circles (Dalla Corte, 1743). It is possible that, in 1488, Giusto began to renovate his country home as a residence located in the middle of nature, which was the perfect representation of one of the ideals of humanist culture at the time [46].

The description of the villa is detailed in a 1495 letter written by Pietro Avogaro (Biblioteca del Seminario Vescovile di Padova, doc. 1652, 647.I), a Veronese lawyer and witness in one of the documents of the legal dispute over the inheritance between Giusto and Zenovello. Over time, this letter has been linked to the Villa Giusti-Melloni [47], later to the Palace of Zenovello [48], and finally to the Villa Giusti-Puttini [31,42]. The author of the letter refers to Giusto, owner of the villa, as follows: "A very illustrious man, Just, a distinguished gentleman among the Just, of whom Lelio was the father, himself a man of the equestrian order, highly sought after in the knowledge of civil law, and admired for all his feats, invited us to his Justianum" (translation by the authors). The handwritten letter, which is 21 pages long, offers a lengthy and detailed description of the villa, called *Justianum*. Special attention was paid to the villa complex, particularly the stately palace, its decoration, frescoes, and even the layout and orientation of the rooms. Regardless of whether it can be connected to the current building, this description depicts a pleasant and wealthy setting for the time, with spaces that, although perhaps small, had elaborate ornamentation and were well-lit, with windows from which the surrounding views could be enjoyed.

After Giusto's death in 1506 (A.S.Vr., Testamenti, mazzo 98, n. 232), his wealth and the title of Count of Gazzo, acquired in 1502, were transferred to his son and sole heir, Gian Francesco Giusti. Gian Francesco Giusti, married to Catterina Serego and later married a second time to Margherita San Bonifacio, was an important figure in Veronese society. In his account of the life of the artist Giovanfrancesco Caroto (Verona, 1480 approx.–1555), Giorgio Vasari talks about count Giovanfrancesco Giusti and his home in Santa María in Stelle, describing a painting by Caroto that the count used as a headboard "in his pleasant

place called Santa Maria in Stelle, near Verona" ([43] p.801). This is not the only reference made by Vasari to the villas of Santa María in Stelle, as the artist Francesco Torbido, il Moro (1562), married the illegitimate daughter of Zenovello Giusti, who gave them an apartment in his palace. Years later, Torbido died in the "beautiful palaces of Santa María in Stelle and buried in the church of that town" ([43], p.804). This is a clear reference to the heirs of the branch of Zenovello Giusti who resided in the New Palace, or Palace of Zenovello. These references to the Giusti family, particularly to Gianfrancesco Giusti, give an idea of how these figures were linked to the cultural and artistic scene of their time.

There is no record of the will of Gian Francesco (deceased in 1544, according to [38]), although there is one for his son Giusto, who drew up his own will in 1560 (A.S.Vr, Testamenti, mazzo 152, n. 684) and, according to a document on inheritance division from 1564 (A.S.Vr, Archivietti Privati, Giusti, Arc. 34, vol. I; ff. 321v–322v), left his property to his son Giusto. The estate of the latter, who died in 1602 with no male heir, was partly transferred to his nephews (the Gazzo property related to the title of Count) and partly to his daughters, the oldest of whom, Isabella, inherited the property in Santa María in Stelle (A.S.Vr, Archivietti Privati, Giusti, Arc. 34, vol. I; ff. 275r–276r). Isabella Giusti married Lodovico San Bonifacio, and in 1622 her son Vinciguerra San Bonifacio inherited the villa from her (A.S.Vr., Testamenti, mazzo 219, n. 655). Upon Vinciguerra's death, this property was transferred to his son Ludovico San Bonifacio, as seen from a valuation of his assets in 1694 (A.S.Vr., TEstimi Provvisori, Polizze d'estimo del 1696, libro II, pág. 347–355, coll. 79). Although no accounts can be found of the villa having been used by Vinciguerra of San Bonifacio and his son Ludovico, a series of drawings suggest that no major transformations were carried out on the complex, so it is possible that it was not in active use. The Sanbonifacio family, an established Veronese family, probably had other properties and may have only paid attention to this villa for the purposes of agricultural activity. The property inherited by Vinciguerra San Bonifacio in 1622 is referenced in the drawing by Ercole Peretti from 1625 (A.S.VE. Beni Inculti, Verona, rot. 74, mazzo 64, dis. 5). This is the first graphic document showing the villa and its elements in great detail (Figure 3).

To this floor plan drawing, which will be analysed below, there followed several axonometric drawings of the complex, showing how few changes there had been in the villa in the space of more than two centuries. The 1683 drawing by Antonio Benoni identifies the complex as being the property of Vinciguerra di San Bonifacio (A.S.VE. Beni Inculti, Verona, rot. 102, mazzo 86, dis. 5); the 1735 drawing by Francesco Cornale identifies it as "Brolo e Palazzo", property of Count Giusto Giusti, son of Giovan Francesco Giusti, with a will dated 1560 (A.S.VR., Archivio Campagna-B. 13, n.154); and the 1798 drawing by Giovan Battista Pallesina identifies it as "Brolo del Conte Maffei" (A.S.VR., Archivio Campagna-B. 32, n. 337). All these drawings were made to document the management of the water distributed to properties and crop fields from the village source. From the 19th century, the property is included in the Napoleonic Cadastre, cited in 1817 as being the property of Uguccione Giusti, son of Antonio, and by the Austrian Cadastre of 1844 to 1886, where it is described as a "holiday home" belonging to the Verdari family, respectively, owned by Gian Battista (1844); Vincenzo and Antonio, brothers of Gian Battista (1849); Antonio, son of Gian Battista (1858); and Battista, Umberto, Cesare, and Rosa; siblings of Antonio (1894). Since 1970, the complex has been owned by the Puttini family.

### 3.2. The Villa and Its Territory

At present, the Villa Giusti-Puttini is made up of the stately palace, the outbuildings for agricultural implements and storage, the dovecote tower, the two courtyards, the garden with a small circular pond, and the orchard and vineyards that surround it. The main street of the village leads to an entrance with rustication, through which an initial courtyard or space is accessed. From there, an open arcade in a volume transversal to the route, which includes the dovecote tower, leads to the entrance to the main space. This second courtyard is delimited on the north and west by the buildings, which in the past were used as a store

for implements and accommodation for the administrator or foreman of the villa; on the third side (south), by the stately palace itself, behind which stands the orchard; and on the fourth side (east), the garden and crop fields are accessed.

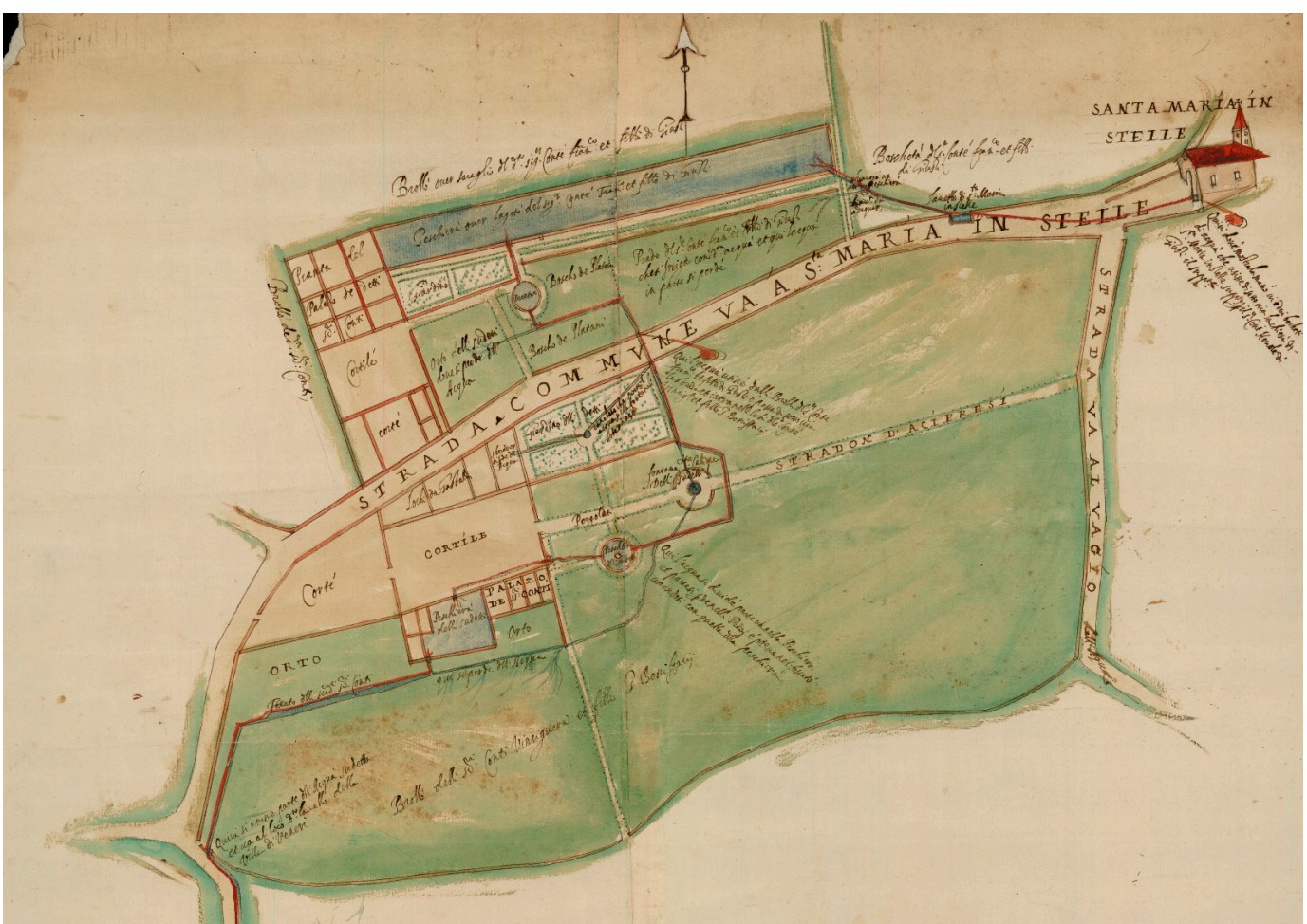

**Figure 3.** Drawing of the complex of the Villa Giusti-Puttini and the Palace of Zenovello Giusti, by Ercole Peretti in 1625 (A.S.VE. Beni Inculti, Verona, rot. 74, mazzo 64, dis. 5).

As already seen, the first clear description of the complex is found in the documents for the contribution of the dowry of Zilia Campagna, widow of Bonsignorio Montagna, whose second marriage, to Lelio Giusti, took place in 1446. The villa is described as "Una petia Terre casaline cum Domibus Muratis, Copatis, et Solaratis cum Curte, et cum Culumbaris et Pischeriis, et cum Broylo cum Arboris fructiferis, et non fructiferis jacens in Villa S. Marie in Stellis". A later description is recorded in a letter by Pietro Avogaro in 1495, describing the complex as follows (summary of the translation by the authors of this text): the enclosure of the villa, called Justianum, located in the middle of the valley with a pond, fountains and gardens is accessed via a stone gate through two courtyards which house the service buildings with kitchen, stores, dining area, oven and the lodgings for slaves and servants; on the other side of the courtyard is the stately palace with the room windows open to the east, overlooking the orchard and the mountains with olive groves, and to the west towards a large pool where fish swim and feed, and with a view of Neptune standing in the middle of the pool.

This description clearly shows that the villa at this time did not differ greatly from the villa recorded on the 1625 plan, identified as the property of Vinciguerra Sanbonifacio. This plan clearly shows the stately palace ("Palazo dei Conti"), the pool ("peschiera") to the left, and to the right, buildings that can be identified as the dovecote-tower, still standing today

in this location. The housing for the foreman ("lochi da gastaldo") was located on the north end of the courtyard ("cortile"). The orchard stood on the south side of the palace, and beyond that, an irrigation ditch, and the park stretched to the east of the palace. Through an opening in a wall to the east, access was gained to the park space ("brollo") from the central courtyard. Along a cross-shaped path covered with a pergola ("pergolata"), a garden with four parterres was reached to the north, while to the south it led to a small circular pool ("peschiera"). To the east, this path led to a fountain, which gave way to a row of cypresses covering the park with pastures ("brollo e pradi") stretching to the village street. The following plans reveal transformations in the complex, although the essence of the villa remains. At present, the first courtyard is surrounded by buildings erected after 1625, as they are not reflected on the plan from this year but can be seen on the 1683 plan, clearly showing the building that is adjacent to the dovecote tower and transversally separates the two access courtyards. The 1683 drawing also shows some houses and a space occupied by the garden and part of the park in the north sector of the property, which are still to be seen on the 1735 and 1798 plans. On the 1735 plan, the pergola covered with climbing plants over the cross-shaped paths between the garden, the pond, and the fountain to the east of the palace can be seen in greater detail. Of all these elements that made up the private garden of the lord and the immediate grounds of the palace, only the small fishpond surrounded with reeds is no longer conserved, although the current paths faithfully follow the original outline.

Compared (Figure 4) with the 1625 plan and those that followed, the garden with parterres has completely disappeared and is now an orchard; the large pool ("peschiera") located between the west façade of the palace and the dovecote-tower was recorded in 1625 as a pond surrounded by walls. This may have been the large pool with fish and a Neptune in the middle and surrounded by arches and columns, as described by Avogaro in his letter from 1495. On the plan from 1683, the same element appears with the name "peschiera rovinata hora horto" (pool in ruins, now an orchard); on the 1735 plan, it is an orchard; and on the 1798 plan, it is a garden. The presence of this pool, later transformed into an orchard or garden, would explain why initially there were no doors providing access to the palace from the west side. However, at a second stage, as recorded on the 1625 plan, an opening was added to connect the palace with the garden on the west side, although no clear evidence of these hypotheses can be found in the structure of the palace.

### 3.3. Fragments Set into the Constructions of the Stately Palace

The stately palace of the villa, located on the south side of the second courtyard, is a seemingly simple building with a rectangular floor plan built predominantly with pebble masonry, lime mortar, and a gable roof. On the main façade, to the north, the main access on the ground floor occupies a central position, to the left of which is a smaller side access, while four arched windows can be seen on the first floor. On the back façade, to the south, are numerous different openings placed at random. On the east façade, it is worth noting a central mullioned window with a balcony, a side window, and a central access on the ground floor. On the west façade, two small doors on the ground floor provide access to the building, and on the first floor, two arched windows light the spaces looking out onto the garden. In actual fact, this seemingly simple layout reveals numerous transformations linked to the different owners and points in time.

Traces of the early owners of the villa can still be found inserted in the palace walls (Figure 5). The Montagna family is present in the form of a stone crest on the main façade and a sgraffito crest in the rendering of one of the two vaulted rooms of the semi-basement, proving that it was built before 1445. There are several Giusti family crests: on the three ledges of the south façade windows, on the painted crests in the small room, and on the crests of the two lunettes painted on the upper section of the west façade. In the small room in the north-west corner of the first floor, the Giusti and Arco family crests are linked by floral elements, symbolising the marriage between Giusto Giusti and Laura D'Arco in 1477,

while the crest of the Serego family represents the nuptials between Gian Francesco Giusti and Catterina Serego in 1502 [28].

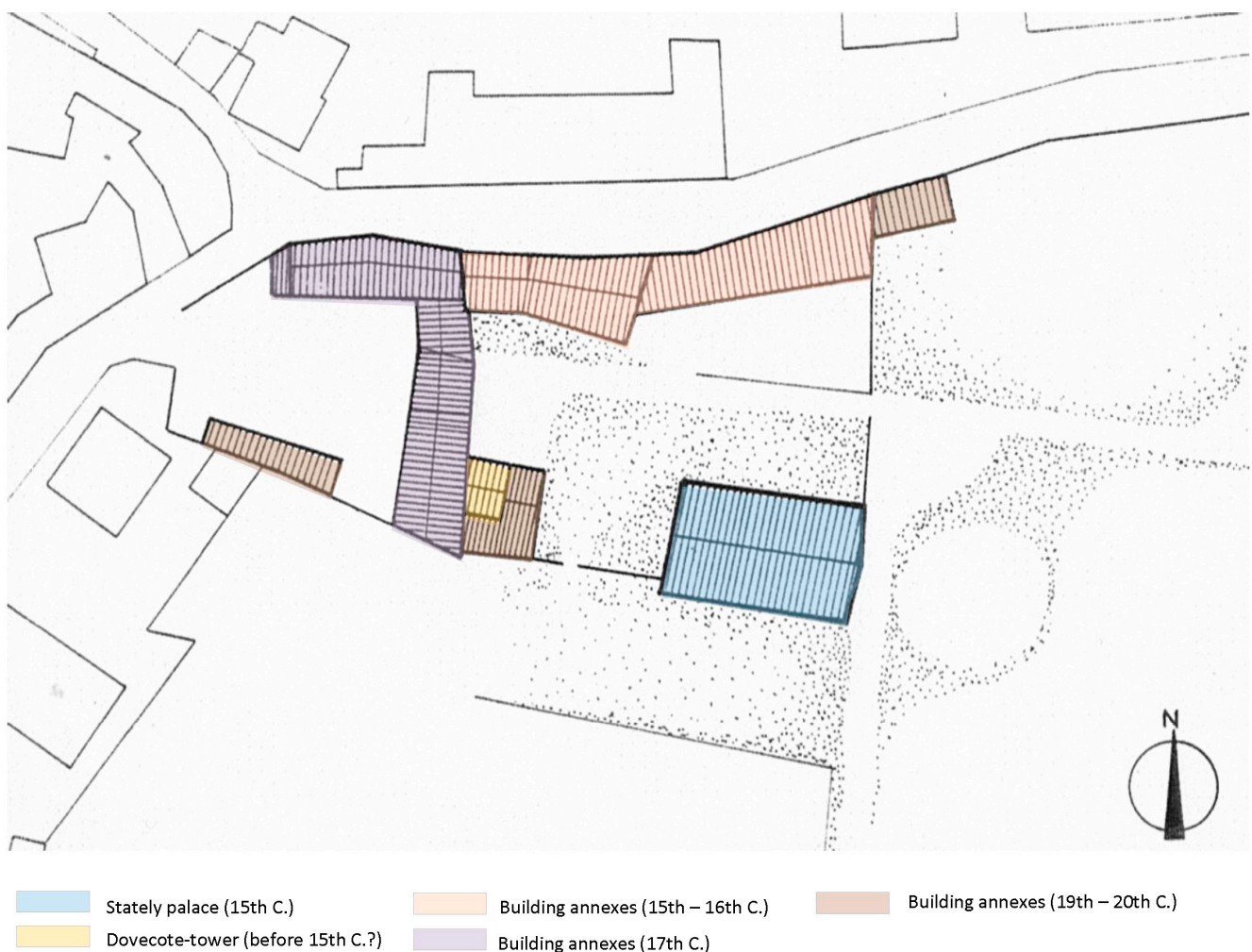

| | Stately palace (15th C.) | | Building annexes (15th – 16th C.) | | Building annexes (19th – 20th C.) |
|---|---|---|---|---|---|
| | Dovecote-tower (before 15th C.?) | | Building annexes (17th C.) | | |

**Figure 4.** The complex of the Villa Giusti-Puttini, showing buildings and garden elements.

In addition to the crests, the palace includes architectural and decorative elements that have been studied both chronotypologically (comparing forms, dimensions, materials, finishings, etc., among them and with similar elements in the same village and province) and stratigraphically, applying the stratigraphical method that allows identifying the temporal relationships between these elements and the building walls. Thanks to the studies carried out, it has been determined that these decorative elements have a clear link to the architecture of the early Veronese Renaissance (Figure 6): the main doorway (B1) and side doorway (B2) of the main façade, the windows on the first floor of the north and east sides (B3), the mullioned window on the west side (B6), the window with a tympanum on the same façade (B5), and the cornerstone of the north-west corner, all feature decorative motifs from the second half of the 15th century. In addition, all the windows on the first floor of the north and east façades are inserted into the construction of pebble masonry, with brick masonry used to adjust the pieces, clearly showing their insertion into a pre-existing construction. In addition to these elements, pieces can be found inserted into successive transformations, expansions, or renovations of the building: the windows on the south façade, from the 16th century, which feature the crest of the Giusti family in the centre of the ledge (C1); the windows with simple yellow stone jambs and lintels, which are on the ground floor of the north, south, and east sides and which were clearly inserted sometime after the 17th century; and the doors of the east side, which most probably date from the

19th century (E). Furthermore, different interventions on the window mouldings can be recognised as imprints of a restoration and remodelling intervention probably dating from the 19th century (B4, B6 . . . ).

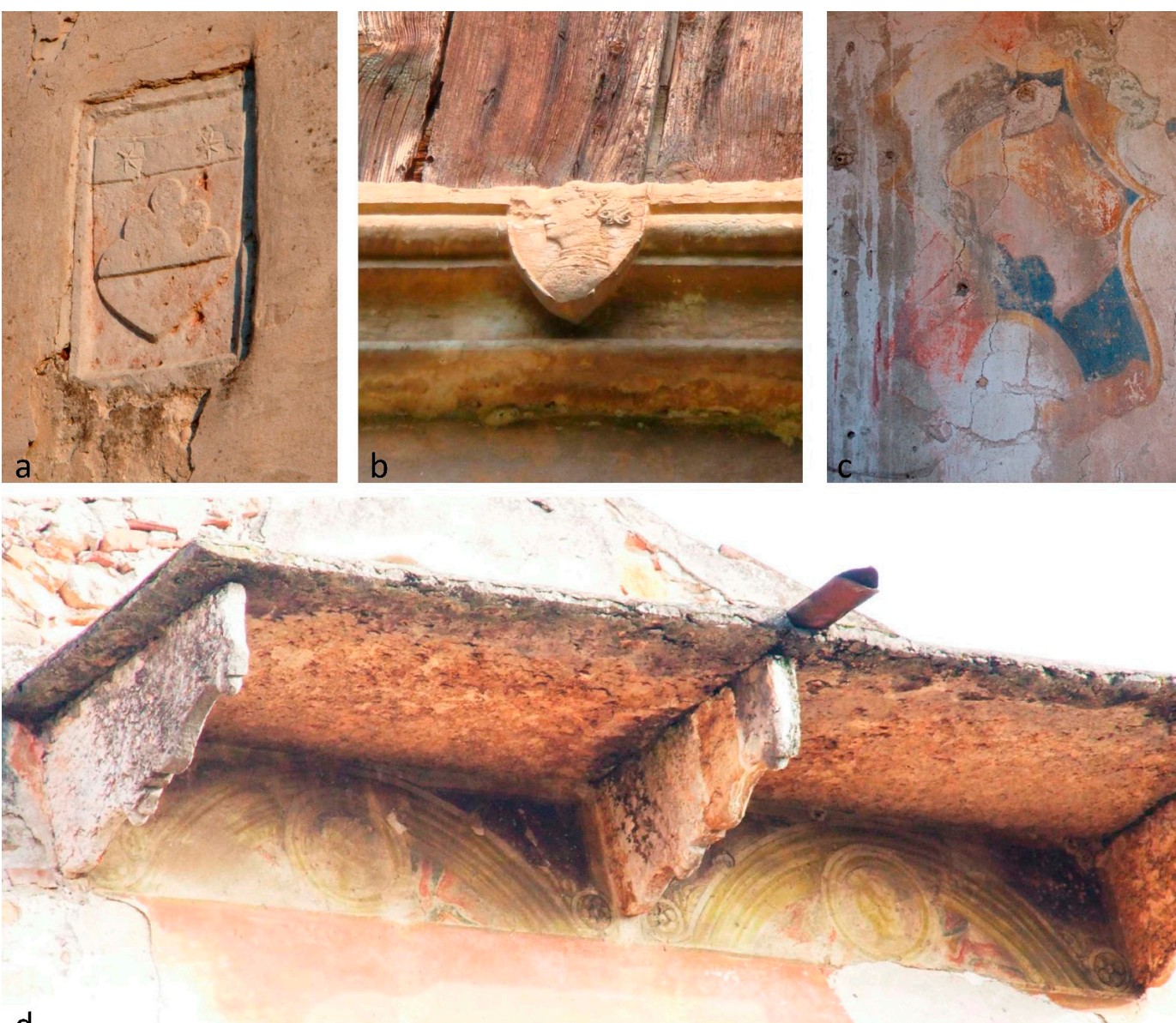

**Figure 5.** Family crests visible in the building: (**a**) crest of the Montagna family and (**b**–**d**) crest of the Giusti family.

On the upper section of the north façade, immediately below the eave, a painted frieze with floral motifs can be made out, which may date from the late 14th to early 16th centuries. However, further detailed study is required of the limited remains of this pictorial element, which may cover the length of the façade, hugging the windows at least to the height of the arches.

It is highly probable that the remodelling work that incorporated the early Renaissance windows (B) may have been started by Giusto when he inherited the palace from his father in 1490. Comparing the mouldings of the doors and windows identified as belonging to this phase with the remains reused from the Palace of Zenovello following its destruction in 1675 and subsequently incorporated into buildings on the same site, a clear correspondence can be observed (Figure 7a). Zenovello began the construction of his palace after he and

his brother reached a settlement on the division of the inheritance in 1490. It is very likely that his brother Giusto started to renovate the family home inherited from his father at this point as well. It therefore seems advisable to compare the windows of the Villa Giusti-Puttini to the windows of the Palazzo Giusti in San Quirico in Verona (current Hotel Accademia) (Figure 7b,c). As stated by other authors [37,38], the transformations of this palace were initiated by Giusto and Zenovello after the death of their father, Lelio, to divide the palace between the two families. These transformations to the city palace could have been executed at the same time as the transformation of the stately palace of Villa Giusti-Puttini by Giusto and the construction of the new palace by Zenovello. The windows of the first floor of the palace in San Quirico are similar in form and decoration to the windows of the first floor (north, east, and west sides) of the Villa Giusti-Puttini (B3, B4). However, it should also be noted that two windows in the villa Giusti-Puttini had projecting iron bars and could be associated with this period. The first of these (B7), on the south façade, corresponds to the side room, and its construction was contemporaneous with the interior frescoes, while the second (C2), on the north façade, underwent a later transformation so that all that remains is the ledge and a jamb, which can be made out in the rendering, showing the imprints of the projecting iron bars. These two windows share many similarities in detail to those found on the ground floor of the Giusti Palace in San Quirico. Finally, the windows with tympanums in the palace can be linked to the window on the stairs (B5) and the side doorway (B2) of the stately palace of Santa Maria in Stelle. The presence of a side balcony in the city palace could also be associated with the corner balcony of the villa Giusti-Puttini. All that remains of this balcony are the cantilever beams surrounding the corner, and the size and railing suggest that modifications were almost certainly carried out on them during the 19th century.

Equally, the passing of time and the detachment of some of the renderings have gradually revealed elements that suggest the existence of doors and windows other than those found currently. Firstly, on the east façade, immediately beside the right-hand side window, the rendering has become detached, revealing the brick jamb of a window located in a wall in pebble masonry. This window (A1) predates the late 15th-century configuration of the first-floor windows and could be linked to the layout observed in the Montagna family home before 1446. Thermography has shown another two elements whose size and shape suggest a possible connection with this window: a window (A2) on the north side and an element on the south side. Whereas the window on the south side can be seen in all the thermographic images, the element on the north façade could be linked to the presence of the small vault on the stair and might therefore not be a window. All the windows and doors of the building feature joinery and ironwork additions from the 19th and early 20th centuries. In some cases, the addition of exterior joinery caused the disappearance of the decoration of the springing of the arches, as in the case of the first-floor windows on the north and east sides.

On the main façade, to the left of the portal principal, the rendering that has become detached shows what is possibly a walled-in pillar (Figure 6(A3)). This element is built with courses of pebble masonry and lime mortar alternating with brick reinforcements (A3). At a height of almost two metres from the ground, there is what may be a brick arch with joints reinforced with lime mortar. This type of joint appears with a bare finish, characteristic of late medieval constructions. The presence of this element was confirmed through thermographic images showing a discontinuity at this point. These remains point to the possibility that this may have been a façade with a portico, a hypothesis supported by different authors at different stages [23,24,31]. Based on the stratigraphic and constructive evidence of this element, the authors of this text have extrapolated the possible geometry of the arch, which, when complete, would align exactly with the axis of the Montagna crest, suggesting a possible connection. A multiplication test was carried out to consider the presence of three or four arches in order to establish the possible existence of a portico, found characteristically in the rural stately mansions of the time such as Casa Montagna in San Zeno di Montagna [41] or Casa Quintarelli in Torbe de Negrar [49], both in the province

of Verona. However, to date, there is no clear evidence of the existence of a portico, which ought to have left more imprints on the lower part of the façade, although obviously the possible presence of this element could be confirmed through tests in the future. For the time being, it is possible to theorise about the existence of a brick arch with pillars in pebble masonry and brick reinforcements, immediately below the crest of the Montagna family. As this arch would have been incompatible with the interior frescoes, it was presumably blocked off before these were painted. In fact, the construction blocking off the arch features pebble masonry very similar to the masonry observed in the original pillars of the same arch, so these were possibly closed before the windows of the upper floor, all of which display brick adjustments, were added.

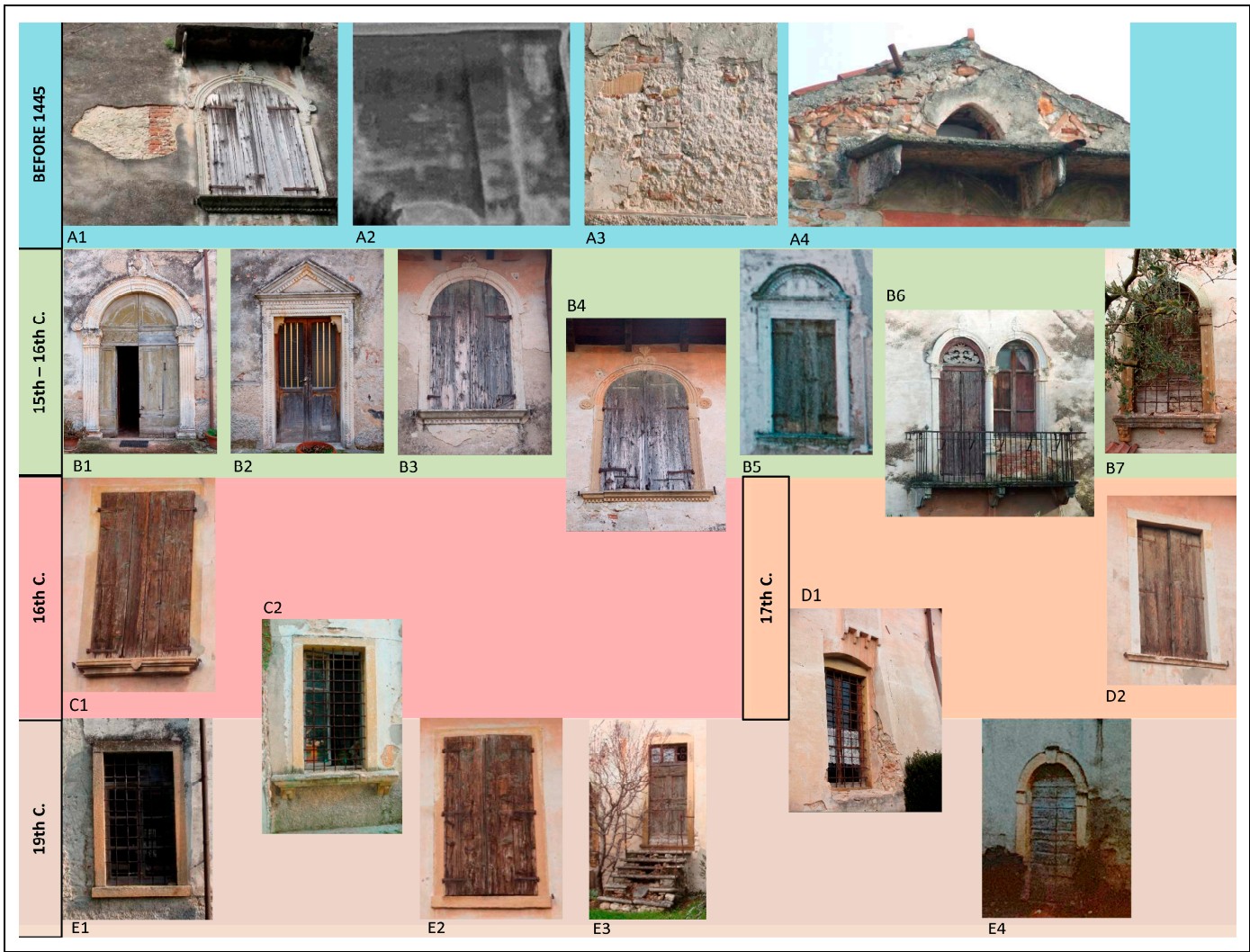

**Figure 6.** Abacus representation of exterior doors and windows of the building in chronological order. The openings have been grouped into four chronological phases (represented by the colored stripes) and have been named with a capital letter corresponding to the period and an identification number.

On the west façade, part of the rendering has become detached to reveal some details of interest. Firstly, there is a clear difference between the construction of the north corner (made with a masonry of small, rounded bolo stones where la torus was added to the corner at the same time) and the rest of the construction of the façade (built with bolo stones, stones, brick, and alternating brick courses). On the upper part of the point of contact, it can clearly be seen how the section connecting to the corner is adjoining some ashlar, which may have been the original corner of the building (Figure 8a). In addition, on the upper part of this face, two painted lunettes with the head of a young man in the

centre (the Giusti family crest) are reminiscent of a mullioned window with the cornices and details of the windows of the north, east, and west faces (Figure 8b). The painted lunettes are protected by a stone bracket similar to those protecting the windows of the east façade. These lunettes are painted on a layer of rendering that covers a series of imprints on the construction: firstly, a small opening with a narrow arch in the centre (also visible from inside the attic) and two small dovecote windows in brick that are now blocked off; secondly, several blocked-off small windows have appeared, especially in the southern part of the façade, which suggest that this part of the façade might have been a dovecote (Figures 7c and 8b). All these blocked-off small windows would suggest that the west front of the palace may have previously been a façade with a dovecote, as observed in more rural constructions or in some stately palaces of the first villas, such as, for instance, the Villa Capra in Carré (Vicenza). Upon inheriting this building, the Giusti family might have wanted to ennoble the west side of the palace, which is visible upon entry into the courtyard. This was probably the reason why the dovecote was blocked off and a fake mullioned window was painted with the Giusti crest and decorative details very similar to those on the window found on the same façade, below, or on other windows on other façades. This operation must also have included the addition of a corner balcony with cantilever beams surrounding the west-south corner of the palace. This balcony was later eliminated, leaving only the current stretch of balcony immediately below the mullioned window.

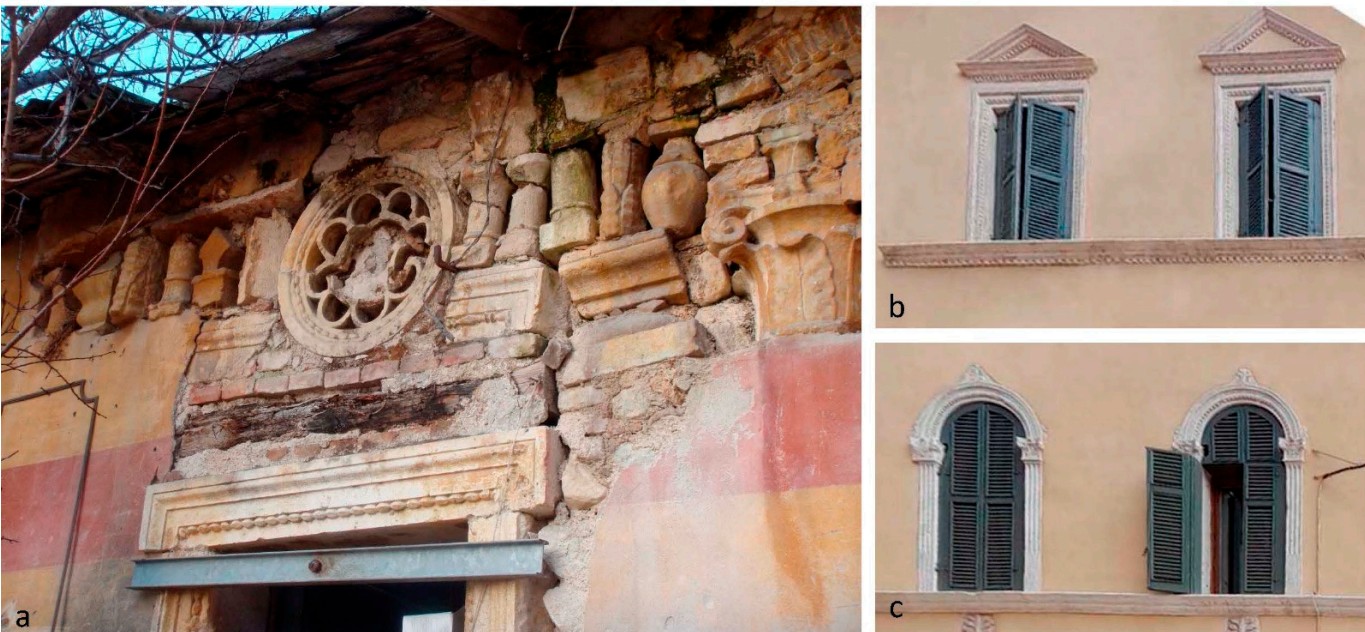

**Figure 7.** (**a**) Fragments of decorative elements of the Palace of Zenovello; (**b**) top-floor windows of the Palace of S. Quirico in Verona; and (**c**) first-floor windows of the Palace of S. Quirico in Verona.

As regards the corners of the building, it should be noted that in addition to the northwest corner with torus moulding in a Renaissance style, ashlar projects from the construction, marking the southwest and southeast corners, both at the base and higher up. Ashlar can similarly be found in other parts of the construction, such as that mentioned in the section between the construction of pebbles and that pre-existing on the west side, or in a discontinuity appearing on the south façade and matching the axis of the first window above the secondary entrance. These elements may have been part of the perimeter of a previous building, which was later expanded. A bulge in the centre of the east façade also suggests the possible existence of a major discontinuity at this point, although further testing of the construction would be required to confirm this hypothesis.

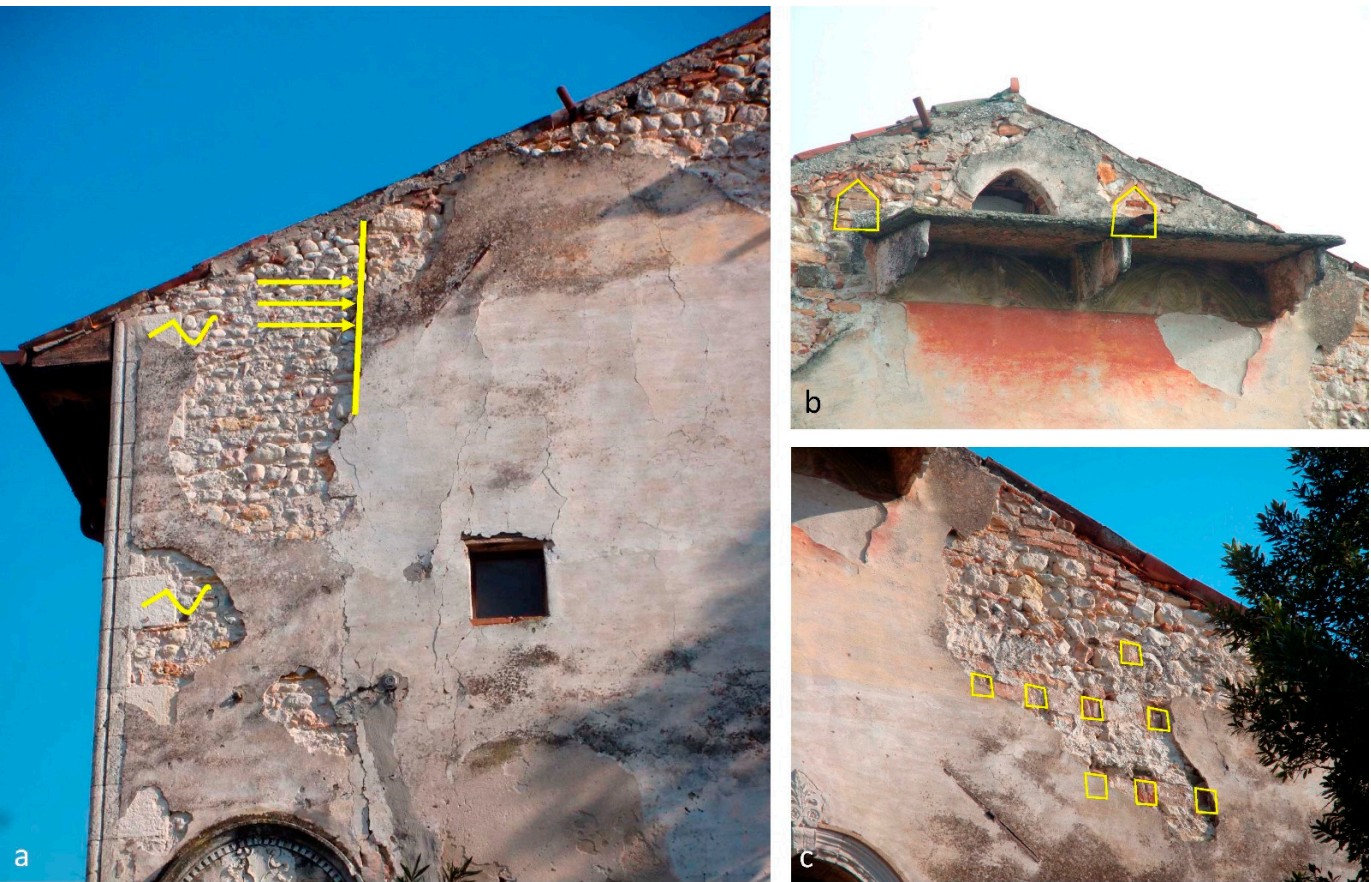

**Figure 8.** Detail of the west façade of the palace: (**a**) the structure of the staircase rests on the former corner of the building: The contemporaneity between the corner and the fabric is visible (highlighted with the symbol ~) and how the same fabric is attached to a pre-existing corner (marked with the symbol ←); (**b**) lunettes of the trompe l'oeil mullioned window and small blocked-off windows; and (**c**) blocked-off windows of the dovecote tower.

The detailed study of the openings and exterior decorative elements is recorded in a scheme (Figure 9), which helps put forward initial hypotheses on constructive periods based on these elements. The information obtained through this study will later be cross-referenced with the data from the study of the structures, architectural elements, and interior decoration of the building in order to present more complete hypotheses in the final part of this text.

### 3.4. The Horizontal Structures

In almost the entire building, the horizontal structures (Figure 10) rely on wooden floors, ceilings, and structures, except in the semi-basement, where there are two rooms with vaults rendered in pebble mortar and other rooms with false ceilings, which prevent the structure from being seen. As the vaults of this semi-basement are built using the same wall technique, they are thought to date from the same constructive period, probably before 1445, given the presence of a Montagna family crest engraved in the rendering.

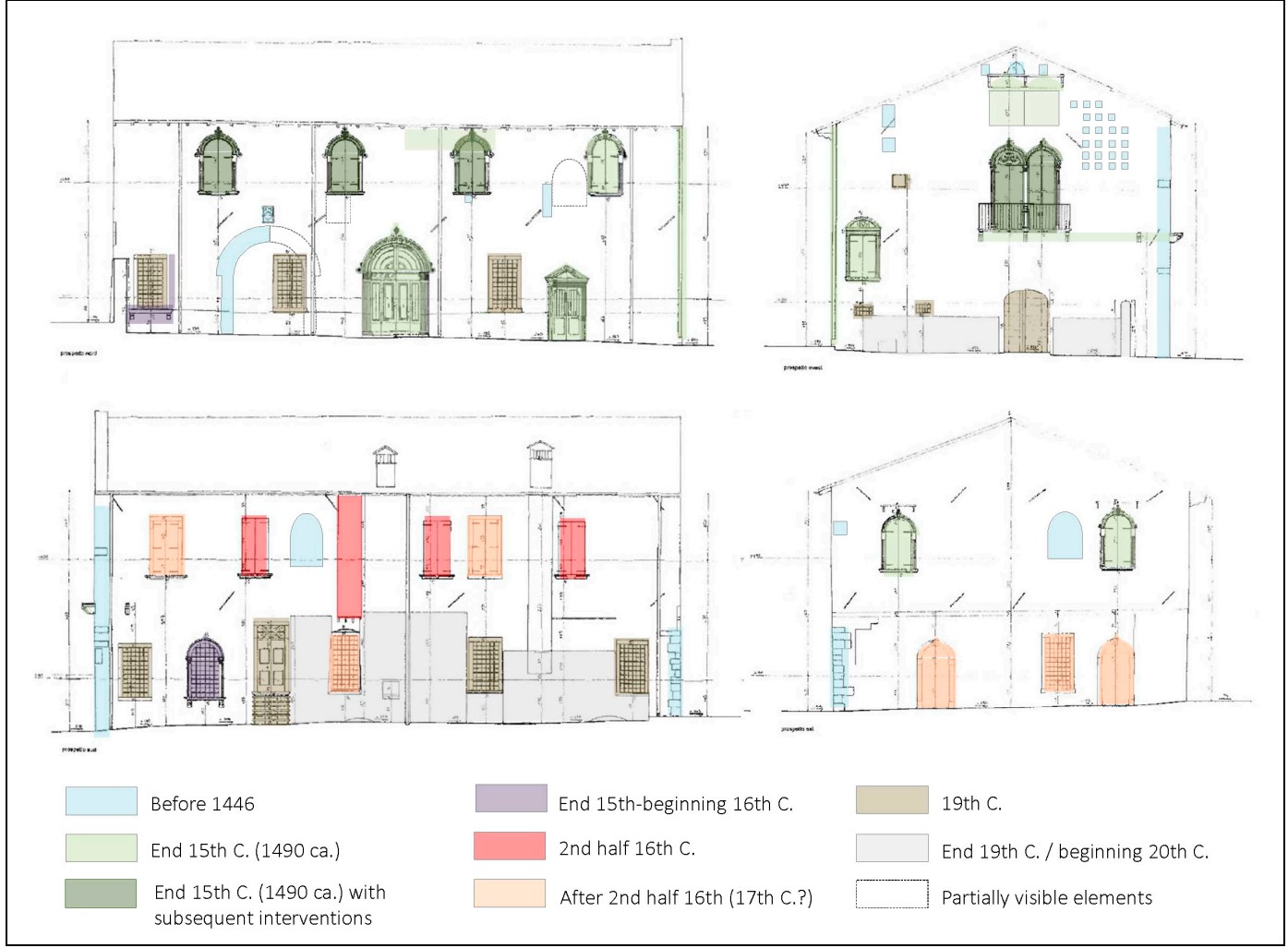

**Figure 9.** Plan showing the chronological order of the exterior decorative elements of the palace.

All the ceilings (Figure 11) are built with main beams, joists, boards, laths between joists, and small boards, except those in the ceilings of the semi-basement (a), where there are no laths or small boards. In some cases, the main beams of the ceiling rest on brackets with very different forms of construction, and in other cases, on wooden wall plates. On the ground floor, the ceiling of the access room (b) and the adjoining one (c) is continuous, and the main beams run from north to south. The ceiling of the current kitchen (d) follows the same north-south direction. However, the ceiling of the room located on the west side (e) runs in an east-west direction. On the upper floor, the ceiling of the room located in the west section (i) repeats the exact layout of the floor below (e), while the rest of the floor plan is covered by a single structure (f-g-h), with main beams running in a north-south direction, which are the tie beams of the roof trusses. These trusses have two king posts and a collar tie on which a post supporting the upper row rests [50]. The ceiling joists of the second-floor rooms rest on the four tie beams of the truss. The lower corners of the tie beams and joists are finished off with tori, which suggests their possible construction in the 15th century [30].

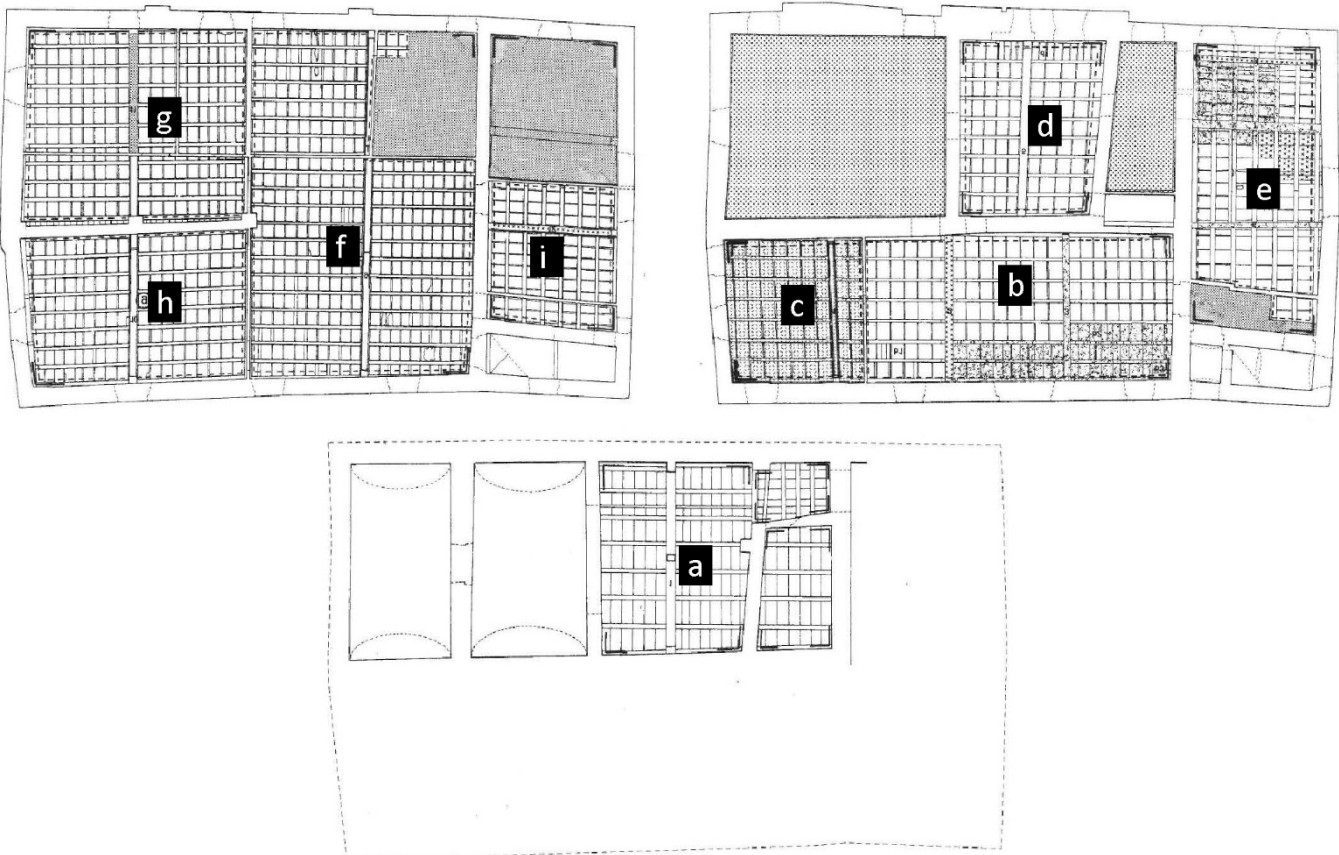

**Figure 10.** Plans of the horizontal structures or floors of the palace (the letters a to i correspond to the various horizontal wooden structures).

Some additional details (Figure 12) could help in the dating process (Figure 13) for these structures given their similarity to typologies and decorations identified in studies carried out around Venice [51]. The laths are observed intermittently on all the ceilings of the palace. This discontinuity was generally common in the second half of the 15th century. These structures differ greatly in terms of decoration and might therefore be the result of different remodelling interventions. Some of them are even clearly repaintings based on other models found in other rooms. In any case, as these structures feature paint on all elements (beams, joists, boards, and laths), the decoration could have been added in the 15th century, when the entire surface was first decorated. Some ceilings display a clearly visible layer in a whitish mix that can be traced to the 16th century: the ceiling of the access room (b), as well as the ceilings of the rooms in the west section of the building on both floors (e–i), are completely covered by a whitish layer on which the pigments of the drawings were applied. Different decoration is added to the preparatory layer in the different rooms: in the access room (b), the beams, joists and laths feature a braided decoration and the boards display a central rosette surrounded by floral motifs, which can be dated to the second half of the 16th century; in the west room of the ground floor (e), the beams and joists (with torus corners) are decorated with interlinked floral motifs and the boards with a circle containing a six-pointed star and surrounded by geometric motifs, perhaps dating from the first half of the 16th century; in the west room of the upper floor (i), the joists are decorated with a floral garland and the boards with alternating floral motifs, perhaps the result of redecoration (second half of the 16th century or first half of the 17th century) of a previous ceiling (second half of the 15th century). On the ground floor, the access room on the north side (b) and the room on the west side (i) contain boards that were reused from previous structures, decorated in very vibrant colours, perhaps reminiscent of the first half of the 15th century. In the room adjoining the access room on the ground floor

(c), the ceiling, which had flowed into the access room (b), was repainted in a continuous cream colour with decoration from the first half of the 19th century.

**Figure 11.** Diagrams and sections of horizontal structures in wood (the letters b to i correspond to the locations according to Figure 10).

However, on the upper floor, the structure covering most of the spaces of the floor plan (f-g-h) has several layers of decoration depending on the rooms: the rooms on the southeast corner (g) appear to conserve their original decoration, with simple geometric drawings combining two colours (black and red and black and white), without a preparatory layer; in the room on the northeast corner (g), a layer with a central rosette surrounded by geometric motifs (rather similar to that found in the access room on the ground floor) was added to this still-visible initial decoration; in the rest of the rooms (f), the beams are decorated with a garland of acanthus leaves tied with red ribbon, while the boards feature a geometric decoration with red and bluish grey motifs against a white background. These joists (even those with a torus on the corner) are 50 cm apart instead of 60 cm, as seen in the structures of the side rooms (g–h). This change in the distance between joists opens the possibility of the previous existence of a single structure (perhaps in the 15th century) and

the possible redistribution of this part of the structure, with some boards changed before a later redecoration stage (perhaps in the second half of the 16th century).

| COD. | Photograph | Laths | Corners | Preparation | Decoration | Interventions | Date |
|------|-----------|-------|---------|-------------|------------|---------------|------|
| b | | discontinuous (15th c.) | Straight | YES | Beams: two–cord braid<br>Joists: two–cord braid<br>Laths: two–cord braid<br>Boards: central rosette and floral motifs | Reused polychrome boards with very vibrant colours and plant motifs can be observed | Structure: 16th c.<br><br>Decoration: 16th c.<br><br>Some boards: 15th c.? |
| c | | discontinuous (15th c.) | Straight | | Beams: cream monochrome<br>Joists: cream monochrome<br>Laths: cream monochrome<br>Boards: cream monochrome and blue stamped motif | The current finish has been painted over the previous ceiling (b) | Structure: 16th c.<br><br>Decoration 19th c. |
| d | | discontinuous (15th c.) | straight | | Beams: some circles can be made out under the soot<br>Joists: some circles can be made out under the soot<br>Laths: cannot be identified<br>Boards: geometric motifs can be made out under the soot | Covered by a thick layer of soot which hampers interpretation | Structure: 15th c.?<br><br>Decoration 16th c.? |
| e | | discontinuous (15th c.) | Torus | YES | Beams:<br>Joists: floral garlands with flowers on opposite sides<br>Laths: cannot be identified<br>Boards: White background with a central star and floral motifs | | Structure: 15th c.?<br><br>Decoration 16th c. |
| f | | discontinuous (15th c.) | Torus | | beams: plan motifs (acanthus leaves with red ribbon?)<br>Joists: plant motifs (acanthus leaves?)<br>Laths: red with white stamped motifs<br>Boards: geometric motifs in red and bluish grey on a white background. Simulation of coffer with two red strips | These were probably painted over h-type monochrome motifs. The boards appear to have been changed | Structure: 15th c?<br><br>Decoration of beams and joists: second half of the 16th c.<br><br>Boards: 19th c. |
| g | | discontinuous (15th c.) | Torus | | Beams: monochrome geometric motifs<br>Joists: monochrome geometric motifs, possibly repainted in red and white<br>Laths: red monochrome<br>Boards: | These were probably painted over h-type monochrome motifs | Structure: 15th c.?<br><br>Decoration: 15th c.? additional paintwork from the 16th and 19th c. |
| h | | discontinuous (15th c.) | Torus | NO | Beams: not visible<br>Joists: monochrome geometric motifs<br>Laths: in uniform red<br>Boards: geometric motifs in two colours black and red / black and white) | | Structure: 15th c?<br><br>Decoration: 15th c? |
| i | | discontinuous (15th c.) | Torus | YES | Beams: not visible<br>Joists: floral garlands<br>Laths:<br>Boards: polychrome geometric and plant motifs on a white background | These were probably painted over a previous ceiling | Structure: 15th c.?<br><br>Decoration 16th – 17th c. |

**Figure 12.** A table showing the constructive and decorative information thanks to which horizontal structures in wood can be dated (the letters b to i correspond to the locations according to Figure 10).

### 3.5. The Interior Rooms over the Centuries

Despite the palace's seemingly simple layout, it is the result of multiple transformations, which are also reflected in the interior. The motifs of the frescoes found in the different rooms have been described in previous publications [19] and studied in detail by other researchers (see especially [28,52]). However, this article aims to take a closer look at the correlation between exterior and interior in terms of the definition of interior rooms, which, in addition to frescoes and paint, were defined by their layout, lighting, flooring, and decoration on walls and ceilings.

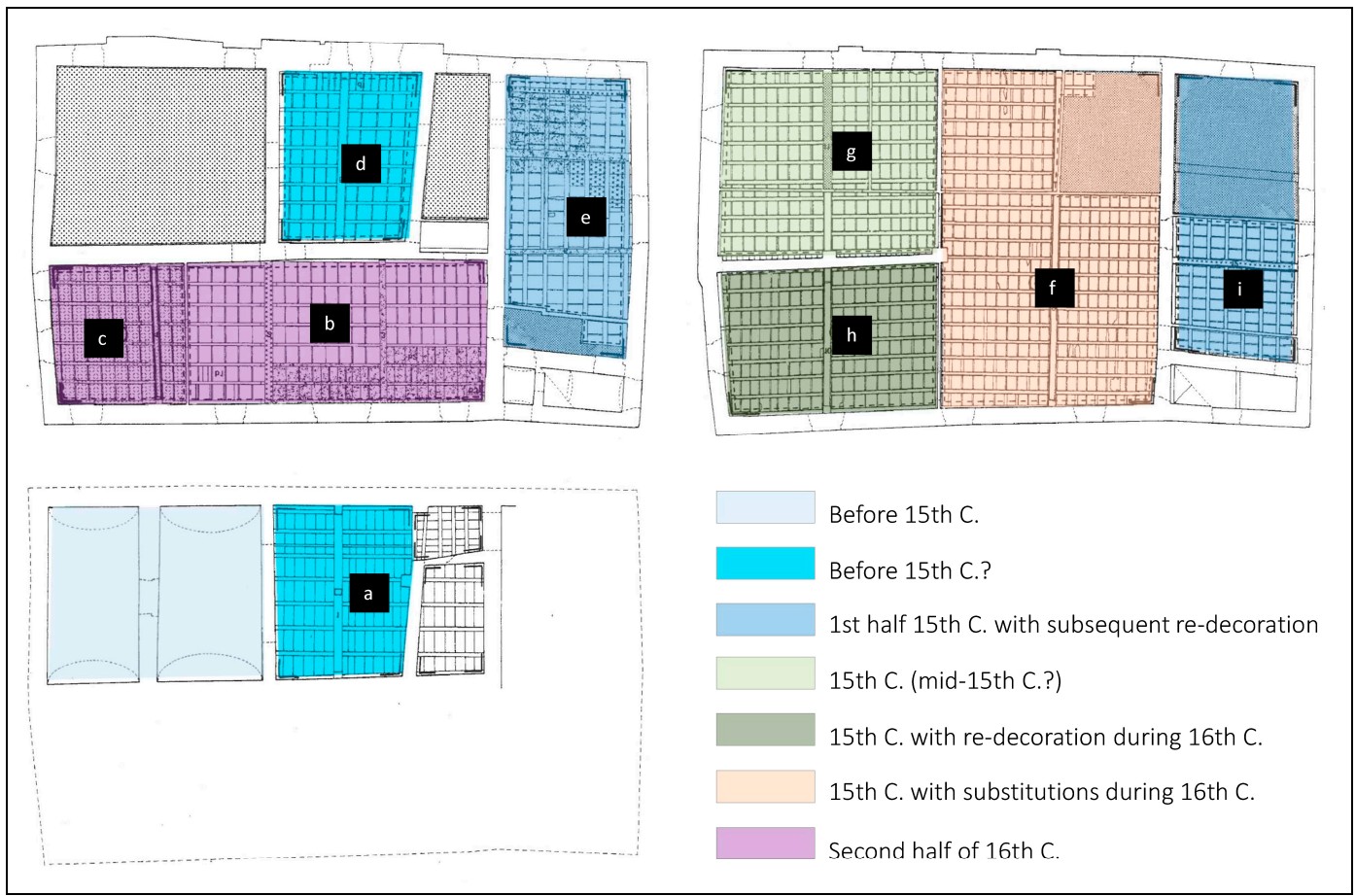

**Figure 13.** Hypothesis for dating the horizontal structures in wood in the palace (the letters a to i correspond to the locations).

The only available description of the complex is that offered by Avogaro in 1495. However, the literary tone and possible transformations undergone by the building since then make it difficult to establish clear connections between the rooms described in the letter and those visible at present. In a summary, Avogaro provides a very detailed description of the palace, beginning with the access rooms: "at the other side of the courtyard stands the palace of the owners, where the work of artists is on display in the access atrium and where the bedrooms, summer and winter rooms all have windows and are decorated with festoons, pavilions, and gold drapes. The most beautiful mural paintings are found in the portico, following the example of Augustus, who was the author of the first ones "with the most pleasant appearance and least expense". The frescoes mentioned depict a crowned Fortune, Diana hunting, and Apollo playing the lyre among the muses. However, these paintings are not the ones that can currently be seen on the painted walls. Avogaro goes on to describe the guest quarters, "which are the most beautiful part and open out onto the pool with arches and columns. From the courtyard, the summer dining room is accessed, with windows that open onto the orchard to the east, overlooking mountains with olive trees; the south-facing winter dining room is decorated with paintings in the classical style and on the corner there is a bedroom that is neither overheated by the sun nor battered by winter winds so that it is comfortable in both winter and summer, with a cupboard in the wall used as a shelf storing books for reading. Between this bedroom and the summer dining room there is a second summer dining room. From there, the open windows provide a view of the large pool in white marble, where the fish swim and are fed, and where in the centre the standing Neptune is on display to spectators. From the side of this dining room, a white marble staircase leads to a more secluded area in the upper portico, where

on a sunny summer day it is already like winter before midday. In the corner of the portico there is a beautiful hall with windows that open onto the orchard and pond. In the most abundant sun there are another two nearby rooms from which the dense forests can be seen, overlooking almost the entire valley."

Despite the transformations, some of the rooms described by Avogaro may still be identified among the rooms now found in the palace (Figure 14). On the ground floor, the two summer dining rooms described could be identified as the access room (B1) and the room occupying the entire west side (B6), while the winter dining room could be the south-facing central room (B4) and the bedroom would be the corner room (B3). The second summer dining room (B6) effectively covered the staircase up to the first floor, as described by Avogaro. Furthermore, on the top floor, the room in the corner of the portico may have been that which now occupies the west side (C9-C10), while the other two rooms located to the south may have been the spaces that are now the two south-facing bedrooms of the central area (C2-C3) and the four rooms of the southeast corner (C5-C6-C7-C8). The successive transformations over time of the interior spaces make their exact identification a highly complex matter.

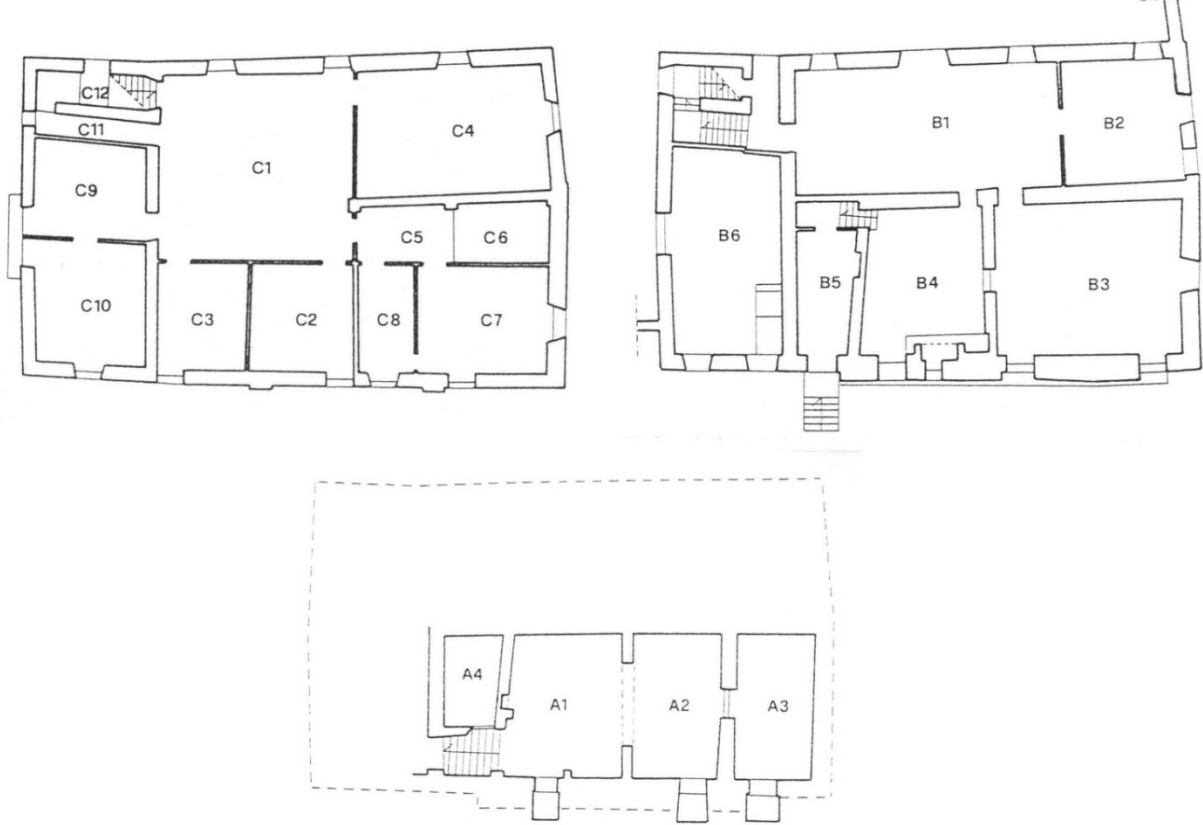

**Figure 14.** Floor plans of all three levels showing room codes (the letters correspond to the different rooms of the palace).

However, there is at least one room still preserving constructive and decorative features that can be associated with rooms from the period in which Giusto Giusti owned the villa. This room is located on the ground floor, where it currently remains as a side room with independent access from the rest of the floor through a doorway incorporated during a later phase (B6). In this space on three walls (the wall separating the room from the staircase is not included as its construction is far more recent), it is possible to find the remains of battle scenes and traces of the sinopias, which could be dated to the late 15th or early 16th centuries (Figure 15). The central section with battle scenes is framed by a plinth that has practically disappeared below and, above, an architectural band with a trompe l'oeil frieze

with clear early Renaissance motifs that continue in the polychrome coffering. The painted frieze must have also had a wooden decoration, now missing. Only one window can be recognised as being contemporary to the frescoes: the easternmost window located in the south wall appears among paintings that cover its jambs and upper arch [30]. Although the deplorable condition of the frescoes (greatly deteriorated due to the damp in a room used for storage and as a cellar) prevents a clear analysis of the iconography, the painted subjects, the trompe l'oeil architectural elements, the movement of the soldiers in classic attire, and the crossing of lances are clearly reminiscent of a period between the late 15th century and the early 16th century. The decoration stretches to the staircase, which was later separated from this room with the construction of a wall. Little remains of the decoration, but it can clearly be seen that it follows the incline of the staircase. The four doors leading onto this space (the secondary doorway for accessing the building from the exterior courtyard, the door leading onto the grand access room, and the two doors by the staircase) all seem to date from the same period and be in keeping with the rest of the decoration of the west room. The ceiling (e) covering the entire room extends to also cover the first stretch of staircase and displays a decoration that is in line with the rest of the room.

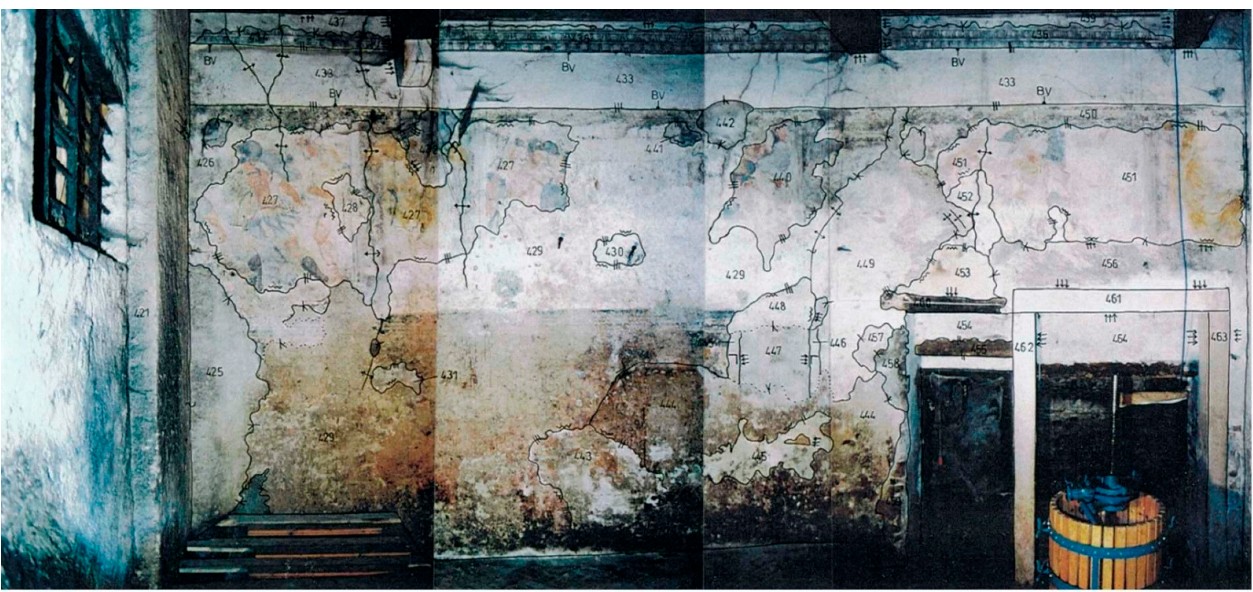

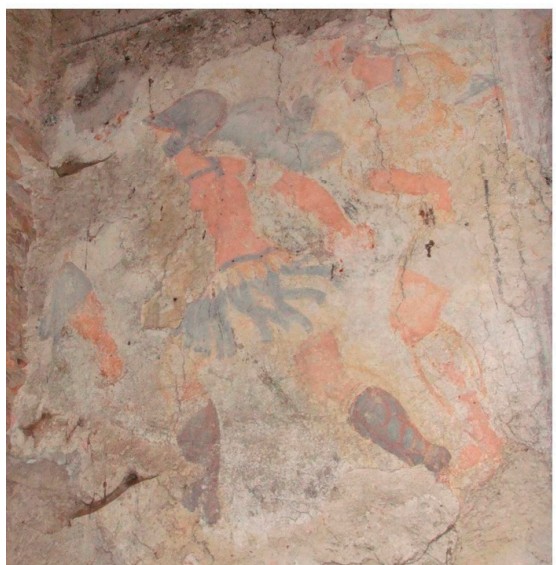
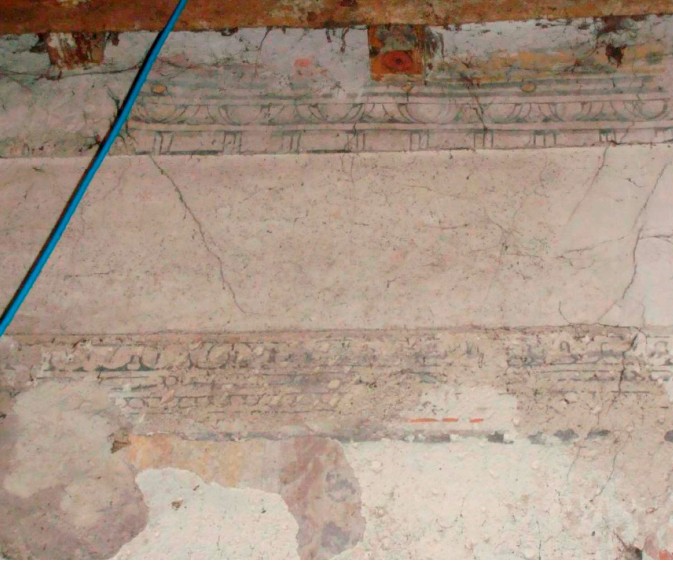

**Figure 15.** General view of the east side of room B6 and detail of frescoes.

The rest of the ground-floor rooms that could be associated with the description by Avogaro subsequently underwent extensive transformations, at least to their surfaces, as the structure of the construction and ceilings could be dated to the late 15th century.

One room that may have existed at that point but whose current layout is clearly more recent is the room for accessing the ground floor (B1). This room, which originally must have been a space for entertaining the public within the palace, conserves a set of Renaissance frescoes on three of its walls (Figure 16). This group of scenes and figures painted between trompe l'oeil architectural elements was clearly created around the mid-16th century. Different authors have attributed these paintings to Veronese authors from this period, such as Bernardino India (approx. 1528–1590) [24], Sigismondo de Stefani (approx. 1525–1574) [28], or Domenico Brusasorci (1515–1567) [23]. The subjects painted between the trompe l'oeil pilasters can be recognised on the end wall as a "Roman triumph" to the left and a "triumph of Venice" in the centre; the side wall features a Perseus and allegorical figure; while a winged victory can be seen on the north wall [28]. These frescoes resemble pictures to be found among the recent limewashes that have been added to much of the pictorial decoration of the room, perhaps due to damage caused by degradation or a series of interventions. The frescoes were a continuation of the decoration of the ceiling (b), as can be seen from the brackets painted on the west wall in imitation of the brackets of real beams. The frescoes continued beyond the wall currently separating this space from the east room (B2), which is now completely limewashed. However, different cuts carried out on all the walls show the presence of frescoes covering the three walls of the room.

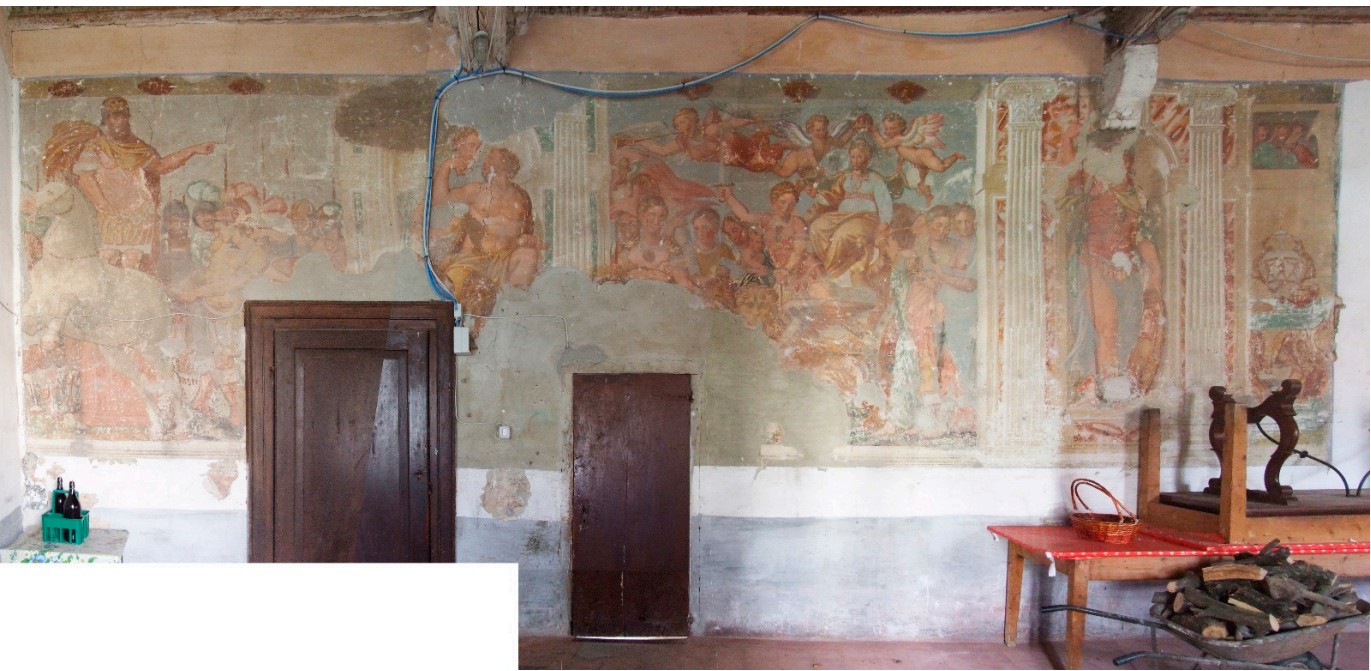

**Figure 16.** Photograph of the south side of the access room (B1) with mid-16th-century frescoes.

In addition, the cuts carried out on the south and west walls have revealed two points of contact that are relevant to the interpretation of the room's layout. In the southwest corner, it seems the west wall continued beyond the current limit created by a wall resting on the fresco. As it is not known whether this was a passage to the back rooms or a niche, this would have to be ascertained with further cuts in the entire perimeter. Furthermore, on the west wall, approximately one metre from the north wall, a discontinuity appears, which must either have been the corner of a niche or of a passage to another space. In addition, the cuts carried out showed that on the west wall, at the level of the lower cornice, there is a lower stratum of frescoes (Figure 17a), probably from the late 15th century to the first half of the 16th century [30], clearly linked to the frescoes and painted cornices

in the west room on the same floor (B6). Finally, it should be noted that, out of the doors and windows currently found in this room, only the door on the west wall appears to be in keeping with the layout of the painted room, while both the doors on the south wall and the windows on the north wall were added after the frescoes were painted. On the west end of the south wall of this space, on the upper section of the painted wall, there are three figures looking out of a window or balcony (Figure 17b). These may have been the patrons who commissioned the paintings. In this respect, the similarity of these three figures to figures appearing in a painting in the National Gallery, originally from the Church of Santa Maria Assunta in Santa María in Stelle, should also be mentioned. In the centre of the painting, attributed to Antonio da Vendri (Verona, 1485?–1545), according to Monicelli's interpretation [38], we find Pier Francesco Giusti, his son Giusto, and one of his brothers-in-law. If indeed these three characters were the same ones painted on the frescoes of the palace of the Villa Giusti-Puttini, these must date from approximately 1540, as Pier Francesco Giusti died in 1544.

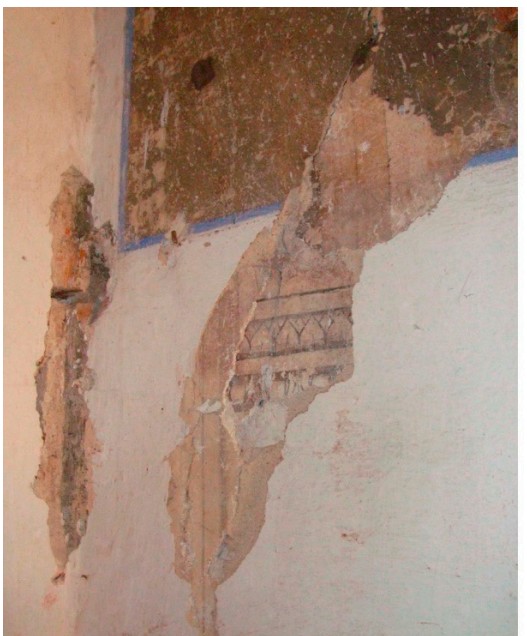 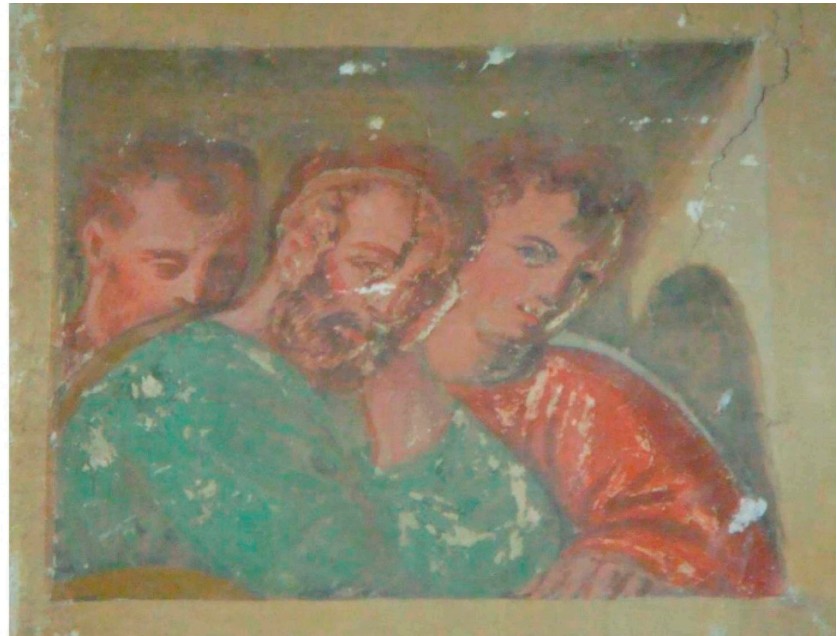

**Figure 17.** Details of the frescoes of the access room (B1 in the map in Figure 14): (**left**), a break in the fresco that reveals the lower layer of fresco painting and (**right**), the three figures on the balcony who are thought to have perhaps been the patrons.

On the second floor, there is a small chamber located on the northwest corner (C12), where there are still some frescoes of interest (Figure 18a). The four walls of this small chamber are painted with a frieze, pilasters, and painted crests joined by garlands. Based on the interpretation of the crests painted in the room (Figure 18b–d), the date of the frescoes has been linked to Giusto Giusti [24] and more recently, Gian Francesco Giusti [28]. In fact, this last interpretation appears to be the most accurate, as the painted architectures show the crests of the Giusti, D'Arco, and Serego families, referring to the marriages of Giusto Giusti and Lucia D'Arco (1477) and Gian Francesco Giusti and Catterina Serego (1503). The paintings therefore date at least from 1506, the year when Gian Francesco inherited the building following the death of his father Giusto. The detailed interpretation of art historians [28] links the decoration to the school of the painter Giovanni María Falconetto (c. 1468–1535) given the use of colours (yellow and red frieze on a blue background; pilasters with grey grotesque motifs against a blue background; and red and yellow capitals) and the decorative motifs (plinth with polychrome marble; pilasters with grotesque motifs; and frieze with plant and animal motifs) (Figure 18e,f).

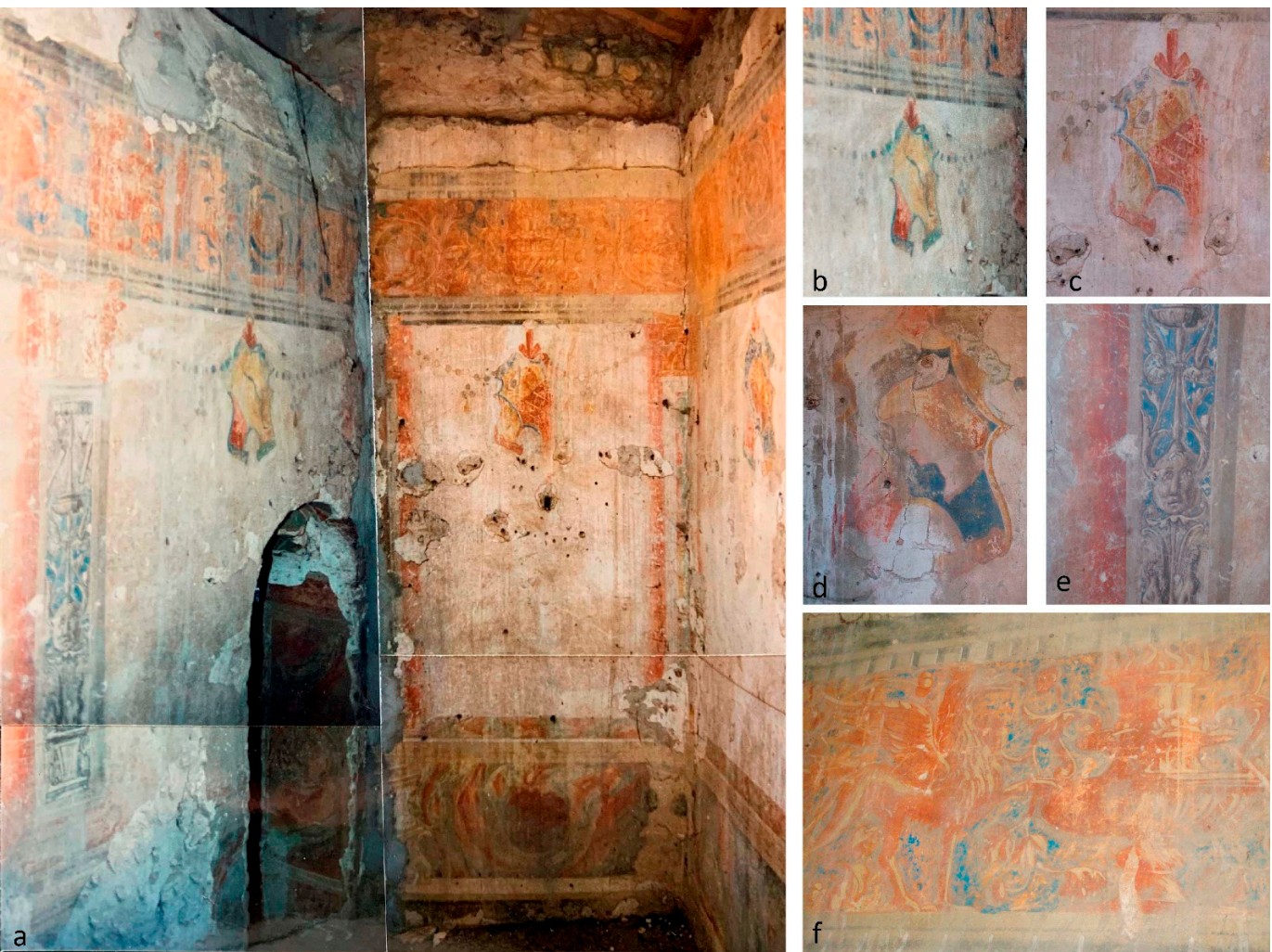

**Figure 18.** Painted small chamber in its entirety (**a**), details of family crests (**b–d**), pilasters (**e**), and the frieze (**f**).

The intrados of the small door used to access the space feature the same trompe l'oeil marble seen on the plinth. The east wall shows the remains of stairs due to the presence of the staircase vault in the section below, which forced the creation of a tall step, with the same marble as the plinth. The decoration of the end wall continues with that of the rest of the walls (the pilasters spring from the plinth, which is exactly the same height as the step), proving the existence of an initial staircase vault at the time the room was painted. However, on the upper section of the wall at the end, there is a break that sheds light on a specific transformation of this space, perhaps in order to gain access to the attic through this small chamber. This may also be linked to the disappearance from this space of the beams, whose imprints are still visible. The construction of a single existing beam suggests a wooden structure may be hidden behind a false ceiling, possibly painted. Finally, it should be noted that the decoration adapts to the window, which is surrounded by the frieze. The intrados of the window arch feature a painted floral decoration with the letters "JU" which could refer either to the name Justo or the surname Justos. However, as the colours and construction of this decoration appear to differ from those in the rest of the room, this panel may have been painted at an earlier stage, perhaps in the late 15th century, at the time of Giusto. It is difficult to know what this space might have been used for; it may have been a small study or perhaps a small chapel located at the side of the main room. Subsequent interventions created a corridor (C11) to provide access to a lavatory, completely cutting off this small chamber from the rest of the house and relegating it to a

secondary role. The main room from which this smaller chamber was perhaps originally accessed is currently divided into three spaces: the corridor leading to the lavatory (C11) and two rooms (C9–C10), probably created in the 19th century. There is now nothing left of this former room. It is likely that the 16th-century room was redecorated in the 17th century, to judge from the decorative motifs that emerge: the painted hunting scene (Figure 19), still visible in the narrow corridor providing access to the smaller room and at one point part of the room wall; the floral motifs painted on the ceiling; and the window opening out onto the south side.

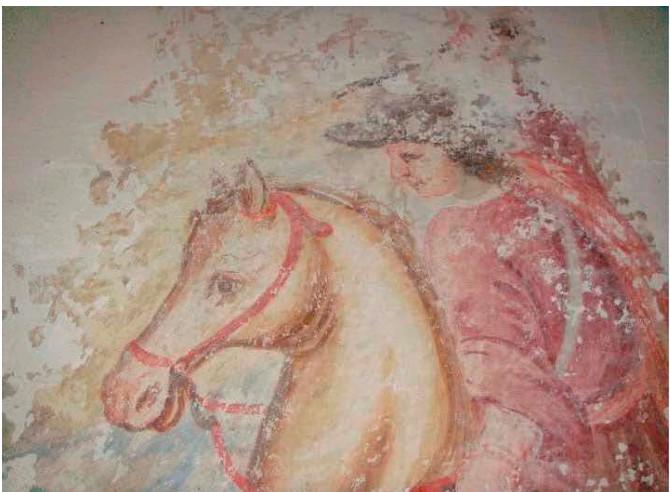 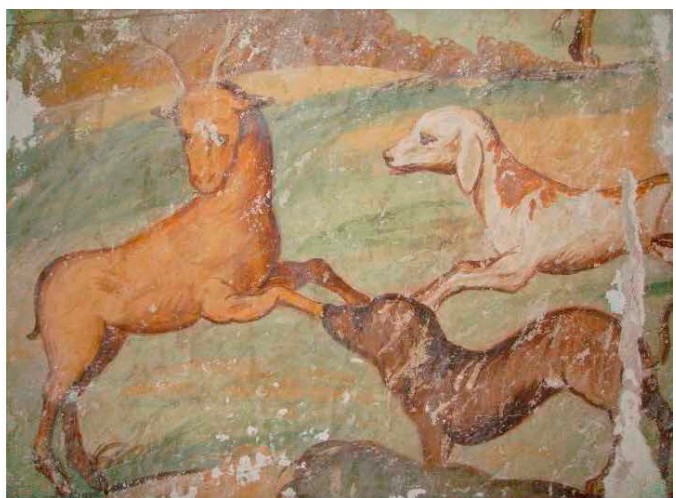

**Figure 19.** Details of the hunting scene painted on the north side of corridor C11.

On the same floor, the grand central hall, divided into three rooms (C1-C2-C3) when it was remodelled in the 19th century, features a series of frescoes with open landscapes between trompe l'oeil architecture. These frescoes have been dated to between the 17th century [24] and the final quarter of the 16th century [28]. The hall occupied the entire central part of the first floor, with windows on both the north and south façades. The remains of the frescoes can currently be seen on three of the four walls that made up this space for public acts in the dwelling. The windows of the north façade (from the 15th century) and those of the south façade (from the 16th century and bearing the Giusti family crest) are surrounded by painted decoration, proving that these were part of the layout of the great central hall. This in turn would confirm that these frescoes were added when the Giusti family was in residence, perhaps in the late 16th century. It is possible that the windows on the north side may have already existed before this hall was created, while those on the south side, which are symmetrical and match the geometry of the decoration, may have been contemporary to the frescoes. On the south side, centred between two windows that show the Giusti crest, there must have been a large fireplace framed between painted marble, from which it is still possible to see the outline of the flue connecting with the exterior of the south façade. This could be the main fireplace, which is conserved in the ground-floor kitchen, or a similar one. It is particularly interesting to note the trompe l'oeil architectural elements framing the landscape views: the stone walls of the west wall of the hall are crowned with shells painted to emulate stone (Figure 20a); framed by finely painted cornices between both doors, we find a landscape with a watermill (Figure 20d), which was broken up when a door was added during a remodelling action in the 19th century, providing access to a lavatory. On the southernmost part of the same west wall, a cut in the current rendering has led to the discovery of a trompe l'oeil door with a frame in sculpted stone and panelled joinery [30] (Figure 20b).

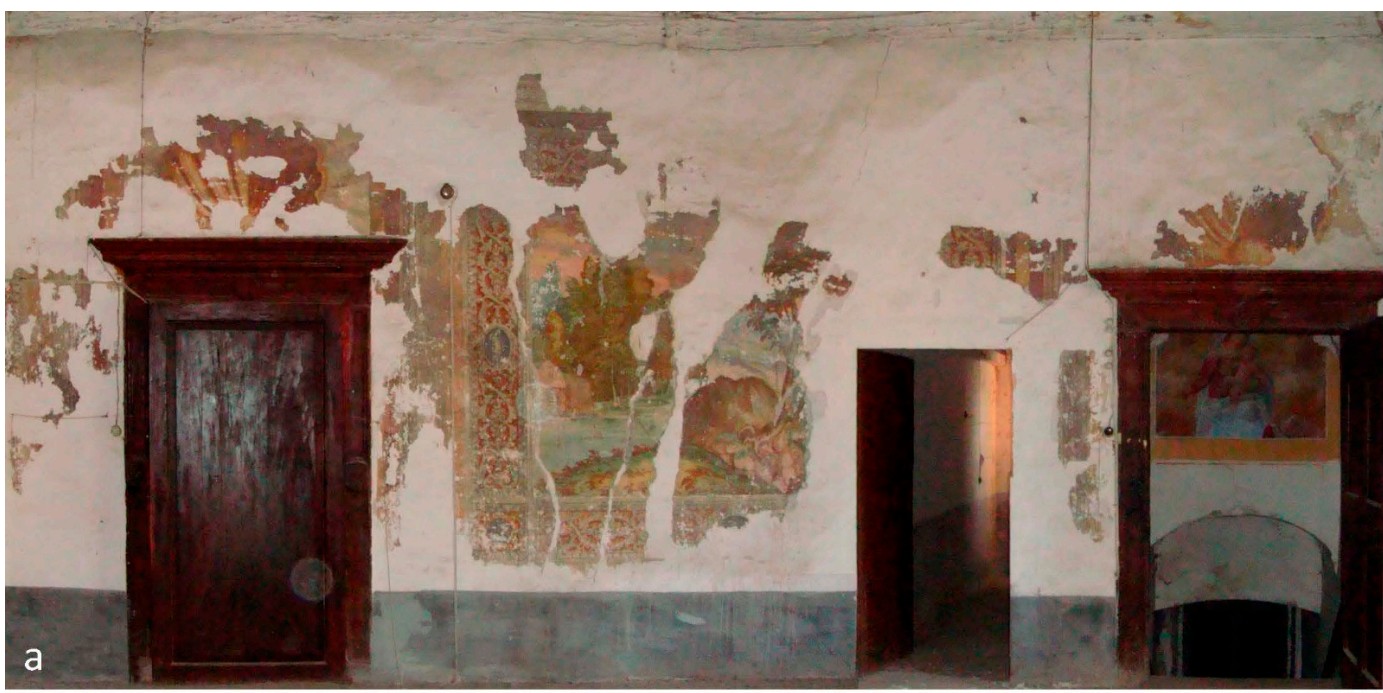

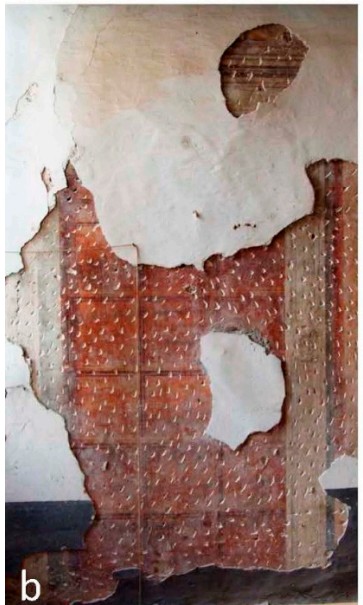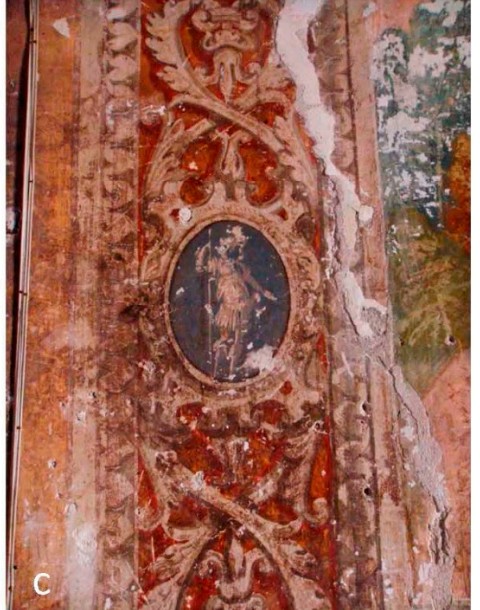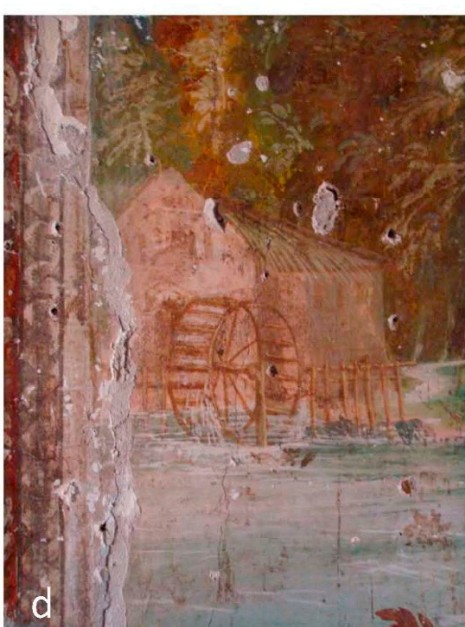

**Figure 20.** Details of the mural paintings and doors of the central hall on the first floor: (**a**) general view of the west wall of the hall; (**b**) detail of the fake door; (**c**) detail of the molding of one of the boxes; (**d**) detail of a watermill painted in a landscape.

The trompe l'oeil door shows the intention of adding emphasis to a central hall with three side rooms, perhaps on both sides, following the tripartite composition of Venetian palaces (Figure 21). In actual fact, the decoration with the composition of three doors in the walls does not match the actual layout of the side spaces but bears witness to the desire to update the most public hall in the palace. The operation for the creation of this central hall must have involved a degree of skill, as the structure of the walls of the ground floors did not correspond to the transversal walls needed for the new layout of the upper floor. It is probable that while the west side of the hall was defined by a pre-existing wall, the east side must have been defined by a lightweight wall made of wood or ceramic tiles, creating the illusion of the central chamber without affecting the structure. This later intervention,

dating perhaps to between the 19th and early 20th centuries, modified the east side of the room, creating pillars on which the tie beam from the truss and some lightweight brick walls rest. It was also probably at this point in time that the wooden doors of the east side were recreated, with wooden frames imitating the stone doors of the west side, simulating the symmetry of the hall.

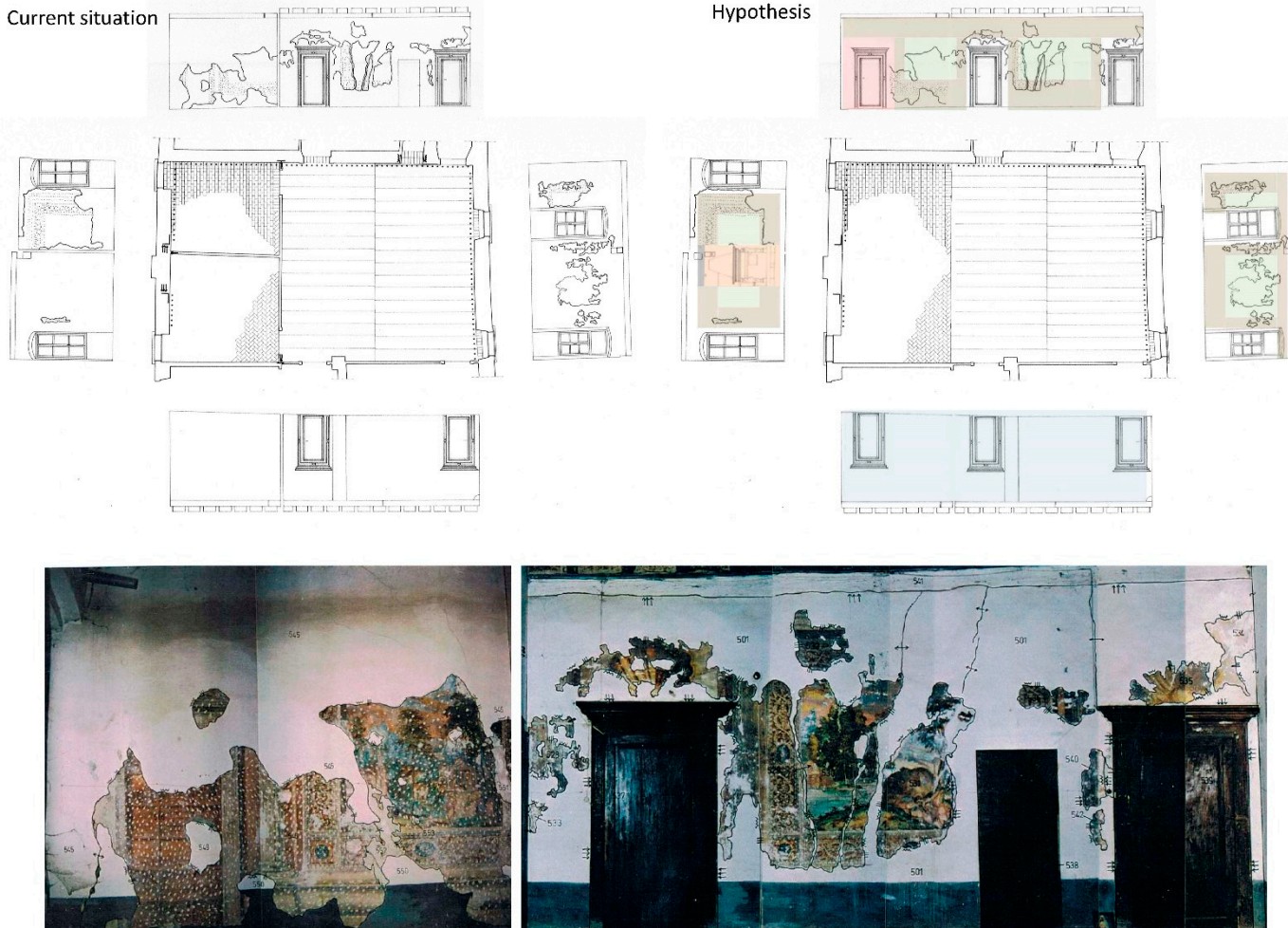

**Figure 21.** Hypothesis for the reconstruction of the central hall on the first floor.

Following the study of the interior rooms and the wall paintings visible throughout, a hypothesis was established for the chronology of the wall decoration, covering all the paintings that are currently visible and the remains of layers that can be seen (Figure 22). The uncovered or visible frescoes, as seen here, mostly belong to the period between the mid-16th century and the early 17th century, a period that almost certainly coincides with the use of the building as a residence where the Giusti family could entertain visitors when living in the countryside.

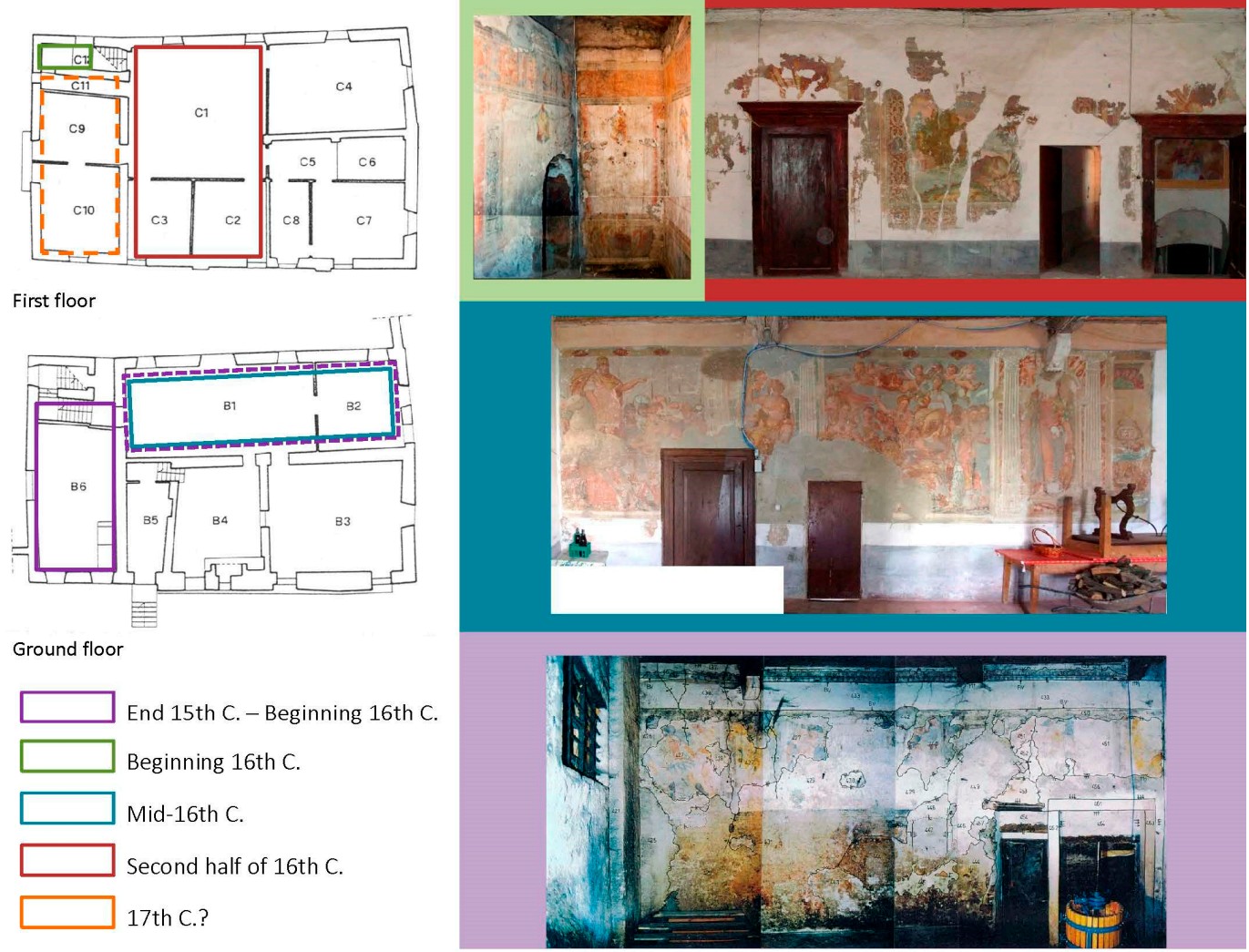

First floor

Ground floor

End 15th C. – Beginning 16th C.

Beginning 16th C.

Mid-16th C.

Second half of 16th C.

17th C.?

**Figure 22.** Chronological hypothesis for the murals in the different rooms of the stately palace. (the letters correspond to the different rooms of the palace).

## 4. Discussion

The detailed study of the historic documentation, of the architectural, constructive, and decorative features of the building, and of the use of analysis methodologies such as stratigraphy and chrono-typology, along with techniques such as thermography, have resulted in more in-depth knowledge of the building and made it possible to put forward hypotheses on the evolution of its construction. As seen above, the hypotheses proposed have focused on the constructive periods of the exterior of the building as well as the horizontal structures and the interior decoration. By cross-referencing these three chronologies, a hypothesis can be put forward in relation to the phases of construction and transformation of the building in all its parts (Figure 23). Obviously, at this stage, this remains a hypothesis as some points are yet to be clarified; in order to do so, further studies would be required, possibly even partially modifying the hypothesis put forward here.

The stately palace of the Villa Giusti-Puttini, as confirmed through historic documentation, predates 1445, when it was passed on from the Montagna family and became part of the estate of the Giusti di San Quirico family, later delle Stelle. Material, constructive, stratigraphic, and typological data suggest that the original building may date to the late 14th or early 15th centuries. In its initial stages, it was possibly a rectangular volume of rustic appearance with two rooms (corresponding to the current B3-B4-B5) on the ground floor and the semi-basement (currently A1-A2-A3-A4) below. In the same period, or per-

haps at a later stage, the two-floor volume on the west side (currently B6-C10-C9) was built. The ashlar with rustication found on the southeast and southwest corners appears to define this perimeter, which perhaps stopped on the east side at the height of the break located exactly halfway along the east façade. At this point, the west façade must have been perforated by the small windows of the dovecote halfway up, with at least one small central window (still visible) built with a pointed arch with bare brick and lime mortar. At the time, this west side was symmetrical in relation to this small window and was delimited to the north by the corner appearing on the upper section of this side of the palace. There are no known documents to support whether the Montagna family already owned the building at this point, although the crest seen in the mortar of the vaulted room of the semi-basement suggests that the Montagna family was responsible for the construction of this part of the building. The characteristics of the construction (thick pebble masonry walls with lime mortar, vaults in lime and pebble mix, etc.) make us think it was a building of some importance, perhaps for rural use or perhaps for use by the aristocracy. It is not known whether the building had a portico on the ground floor or a portico on the ground floor along with a corridor or gallery on the first floor, perhaps both in timber, according to the hypotheses relating to other buildings of the time, such as Casa Montagna [41] or Casa Quintarelli [49]. However, there are no clear imprints of these elements at present.

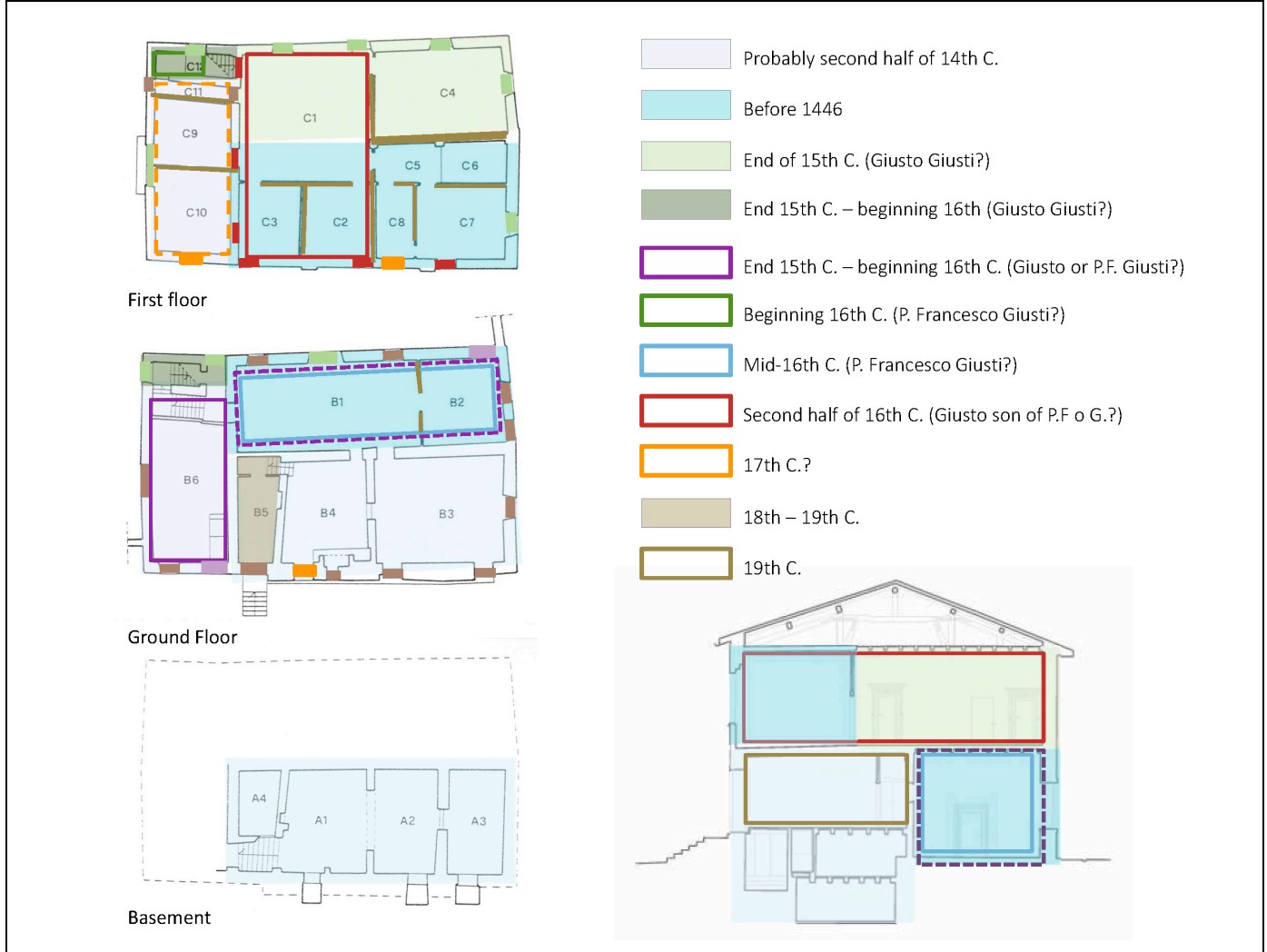

**Figure 23.** Hypothesis for the phases of construction and transformation of the stately palace of Villa Giusti-Puttini (the letters correspond to the different rooms of the palace).

The following transformation may well have taken place in the first half of the 14th century when the building was expanded from a bay oriented to the north and what is now the access room on the ground floor (B1) was added, while the top floor was extended to the same point, creating the north façade. This operation resulted in a building volume very similar to the one found currently. It was probably at this point that the Montagna crest was placed on the north façade of the building, as well as the blocked-off access arch identified below the crest on the same façade and the blocked-off window that can be made out under the rendering of the east façade. It was possibly also in this transformation phase that the wooden structure of the trusses and ceilings of the second floor were added, although it may also date from the constructive phases after 1490, which can be linked to Giusto Giusti's acquisition of the villa. This third phase corresponds with the point at which attempts were made to transform the building into a Renaissance palace by altering its appearance, especially on the north, east, and west sides, by adding the central access doorway, the secondary doorway, the first-floor windows, and the frescoes from the interior rooms. The frescoes and the decoration of the ceiling of the ground floor (B6) and the lower layer of the frescoes found in the entrance hall on the ground floor (B1) may also belong to this phase. It is at this point that the brick walls that surround the staircase and the small chamber on the second floor must also have been added, perhaps transforming a previous staircase or building a new one. It is thought that the mullioned window with the Giusti crests on the upper part of the west façade, painted to ennoble this side of the palace, could also date from the same period or the one that followed. This is followed by an operation that can be dated to the early 16th century: the decoration of the small room, ordered by Gian Francesco Giusti in 1506. He is perhaps also to thank for the frescoes painted in the mid-16th century in the hall of the ground floor (B1) and the replacement and complete redecoration of part of the ceiling.

The following transformation phase, around the second half of the 16th century, affects the adaptation of the hall in the central part of the first floor. This transformation entailed the opening in the south façade of three windows with the Giusti family crest in the centre of the ledge, the insertion of the doors on the west side of the hall, and the decoration of the frescoes of the entire central hall in the Venetian style. This was clearly an intervention to remodel and update what was a stately palace from the second half of the 15th century or the first half of the 16th century to fit the trends of the time. It was probably an east-west composition with two main longitudinal halls superimposed and oriented to the north façade and rooms that opened to the south from there, with a tripartite composition, a central chamber in a north-south direction, and side rooms [19]. Following this update to the building, which indicates a degree of interest in its being used as a building where the family could entertain, there is only the trace of another intervention that was possibly carried out in the 17th century in the west room of the first floor (C9-C10-C11), where a new south-facing window was added, the ceiling was redecorated with floral motifs, and walls were painted, perhaps all over the room. All that remains of this layer of paintings is a small detail from a hunting scene (C11).

The building layer underwent a major intervention in the 19th century, possibly in the second half, when the palace was used as a summer residence by the Verdari family. It was then that the current layout was established in the palace, better suited to the demands of daily life than to representation. Several details suggest that the owners invested in remodelling, adapting this building to new needs. For security reasons square iron bars were installed on all the ground floor windows; as glazing was available, interior and exterior glazed joinery was added to all the windows; on the first floor, the great central hall or chamber and large side rooms were divided using lightweight walls in brick or reed to create smaller rooms that could be made better use of; another room was created on the ground floor (B2) by inserting the wall separating the side room from the access room; on the ground floor entrance doors were added on the east side and finished off with their exterior and interior joinery; the flooring throughout the house was replaced with ceramic flooring in a herringbone pattern clearly adapted to the new rooms; services

such as the kitchen were adapted; on the top floor a lavatory was installed (C11) eventually separating the smaller chamber from the room to the point that it was left in total disuse, as seen from the lack of interior joinery. Interior doors were added, both on the upper floor, where an attempt was made to maintain the symmetry of the main chamber, and on the ground floor, where the frescoes were broken up with the access hall connecting with the kitchen and dining room; the walls were limewashed, covering the frescoes, and the old ceilings were repainted or concealed under hidden ceilings. To these interventions, we can add others carried out in the 20th century: the replacement of flooring in the ground floor access room (B1), the addition of a cabin structure by the main doorway, the insertion of some new doors; the intervention to repair the roof, replacing all the boards and reinforcing the trusses with new structural pieces or with the construction of the pillars on the first floor, reducing the span of the tie beams; the creation of access to the space below the roof from the small chamber, etc.

## 5. Conclusions

The study carried out on the Villa Giusti-Puttini has resorted to a combined multiple methodology of indirect sources (archive documents, iconographic sources, and documentary sources) and direct sources (the grounds of the villa and the stately palace), and disciplines such as history of architecture, construction, and art, constructive and structural analysis, archaeological and chrono-typological observations, etc. In addition, different techniques were used for metric surveys, photographic documentation, stratigraphic studies, thermographic studies, etc. All these cross-referenced studies combined have resulted in extensive knowledge of the complex.

This text has attempted to show how a detailed study methodology with different types of interpretation that can be interlinked can result in a credible hypothesis based on real data. The authors have endeavoured to show how, in the field of history of architecture as well, the direct interpretation of the building provides the necessary data for the correct interpretation of the building's constructive phases. In this regard, this article provides a study methodology that transcends the local setting and could be applied to research the history of architecture, the history of art, and the history of construction.

The detailed studies carried out in the stately palace have shown it is possible to interpret its constructive and architectural evolution through verifiable material data. The interpretation of historic buildings is often based on stylistic, typological, or historical hypotheses. However, the study of material data helps base these hypotheses on incontrovertible material evidence. In the case of the stately palace of Villa Giusti-Puttini, there are still questions to be answered, as some rooms are completely covered in thick rendering, while false ceilings mean that the number of elements that can be studied is very limited. However, thanks to the study of decorative elements, wooden structures, and interior rooms, an overall hypothesis was established that overcomes most of these difficulties. These elements, studied in detail and collected using abacuses and chronologies, follow a chrono-typological basis that can be applied locally and regionally to interpret and date other buildings from between the late Middle Ages and the Renaissance.

In addition, this article provides a new interpretation of the history of architecture. Architecture evolves and transforms based on the needs created by use, fashion, and the representations required by owners and users over time. Therefore, it is almost impossible to classify a given building at a specific time in history. Every building is the result of multiple transformations. More can be learned about the history of architecture from these transformations than from a building seen as static and fossilised in time. The owners' desire to transform and update this building clearly reflects their needs, interests, and tastes while also shedding light on the specific types of technical intervention needed to carry out this transformation. In the case of the stately palace of Villa Giusti-Puttini, both academics and the general public are confronted with a building that can show various stages of the history of architecture of the villa in the province of Verona and, more generally, that of the villa in the Veneto. This article has contributed to clarifying some of the characteristics of

this phenomenon, which are relevant, not only to the history of art and local architecture but also at a national and international level, given the influence that the models of the palaces and villas, particularly the Palladian ones, had in many countries in Europe and beyond.

To sum up, the research described in this article has contributed advances applicable beyond the complex studied, potentially influencing at both a national and international level: it offers an example of multiple methodologies that can further knowledge of a specific building through the interaction of different disciplines, methods, and techniques; it provides a basic chrono-typology of decorative, constructive, and structural elements that can be the basis for interpreting similar or contemporary buildings from the same area or from the Veneto. Finally, it shows a history of complex architecture that is not frozen in time but based on the transformation of the buildings.

### Documents and Archives

Archivo Nacional de Venecia (A.S.VE.):

- A.S.VE., *Beni Inculti—Verona*, rot. 102, mazzo 86, dis. 5
- A.S.VE., sez. *Fotoriproduzioni—Beni Inculti, Verona*—rot. 74, mazzo 64, dis. 5

Archivo Nacional de Verona (A.S.VR.):

- A.S.VR., *Archivietti Privati*, *Giusti*, Arch. 34, vol. I; hojas 213r-213v
- A.S.VR., *Archivietti Privati*, *Giusti*, Arch. 34, vol. I; hojas 27r-29v
- A.S.VR., *Archivietti Privati*, *Giusti*, Arch. 34, vol. II; hojas 19r-22r
- A.S.VR., *Archivio Campagna*, B. 13, n. 154
- A.S.VR., *Archivio Campagna*, B. 13, n. 155
- A.S.VR., *Archivio Campagna*, B. 13, n. 337
- A.S.VR., *Archivio Campagna*, B. 32, n. 337
- A.S.VR., Catastro Austriaco—Santa Maria in Stelle, funda 61

Biblioteca del Seminario Vescovile di Padova (B.S.V.PD.)

- B.S.V.PD., doc. 1652, 647.I (author: Pietro Avogaro; year: 1495)

**Author Contributions:** Conceptualisation: C.M.; methodology, C.M. and F.V.; documental research: C.M.; on-site research: C.M. and F.V.; photos: C.M. and F.V.; drawings: C.M.; writing—original draft preparation: C.M.; writing—review and editing, C.M. and F.V. All authors have read and agreed to the published version of the manuscript.

**Funding:** This research was self-funded by the authors.

**Data Availability Statement:** Not applicable.

**Acknowledgments:** This research was made possible thanks to the kindness of the owners of the villa Giusti-Puttini, who readily allowed the authors access to the building at all times. The thermographic study was carried out in collaboration with José Luis Lerma García from the Universitat Politècnica de València.

**Conflicts of Interest:** The authors declare no conflict of interest.

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
