# Peer review of "Fragments for the History of an Architecture: A Villa between Humanism and the Renaissance"

_2673-8945, doi:10.3390/architecture3030020_

Round 1

Reviewer 1 Report

Dear authors,

the contribute is well structured and clearly developed. Only a few suggestions to make it more effective and emend some tiny formal aspects:

 -1Lines from 410-426 – I’d suggest to clarify in the text or in the caption of fig.5 the criteria (cross-reference between chronotypology and stratigraphy) allowing dating the elements described in the table.

-From line 637 (paraphrase): maybe the orginal text could be put in a note?

 -A quick review would help emend double spaces and check if the dates of the references inside the text match the ones in the final bibliography

 -Formal remarks: fonts of the text should be checked (from line 269 to 324 and final references have a different font or mixed fonts)

 Best wishes

Author Response

Thank you very much for your suggestions that have been taken into account as follows:

-Lines from 410-426 – I’d suggest to clarify in the text or in the caption of fig.5 the criteria (cross-reference between chronotypology and stratigraphy) allowing dating the elements described in the table: DONE

-From line 637 (paraphrase): maybe the orginal text could be putin a note?

The letter is made up of 23 handwritten pages where the description of the house is mixed with mythological stories and other types of considerations unrelated to the house, so the purpose of this article is not considered to be the full transcription of the document. The phrase “paraphrasing the original letter” has been eliminated and the literal translation of some paragraphs has simply been left.

-A quick review would help emend double spaces and check if the dates of the references inside the text match the ones in the final bibliography: DONE

-Formal remarks: fonts of the text should be checked (from line269 to 324 and final references have a different font or mixedfonts): DONE

Reviewer 2 Report

Please check the character body in references and the quality of images

Author Response

Thank you very much for your suggestions that have been taken into account as follows:

Please check the character body in references and the quality of images: DONE

Reviewer 3 Report

This is a well-organized and properly written manuscript that fits nicely within the Architectural Heritage scope of Architecture, which in my opinion would be of great interest for the target audience of the journal. In it, the authors presented a detailed study of the Villa Giusti-Puttini in Italy. The authors have followed a methodology that combines direct and indirect sources of information and managed to extend the scope of previous studies in the topic. In my opinion, the manuscript can be published in Architecture after the authors attend the following comments/suggestions:

Major comments:

·         There are no major comments.

Specific comments

·         In the introduction, include in a short sentence the meaning/definition of “villeggiatura”.

·         Include a map with the geographical location and extension of the discussed region in the paper (Veneto) and highlighting the location of Villa Giusti-Puttini on it.

·         There is a change of font type between lines 269 and 324 (this issue must be fixed perhaps by the editorial services of the journal before publication).

·         Increase the contrast of colours used in Figures 3, 8, 12 and 22 to indicate the different building periods to improve its accessibility and facilitate its interpretation.

Author Response

Thank you very much for your suggestions that have been taken into account as follow:

- In the introduction, include in a short sentence themeaning/definition of “villeggiatura”: DONE

- Include a map with the geographical location and extension ofthe discussed region in the paper (Veneto) and highlighting thelocation of Villa Giusti-Puttini on it: DONE

-There is a change of font type between lines 269 and 324 (this issue must be fixed perhaps by the editorial services of the journal before publication): DONE

-Increase the contrast of colours used in Figures 3, 8, 12 and 22 to indicate the different building periods to improve its accessibility and facilitate its interpretation: DONE